# Do Large Language Models Truly Understand Geometric Structures?

**Xiaofeng Wang, Yiming Wang, Wenhong Zhu, Rui Wang***
Shanghai Jiao Tong University
{banyedy, yiming.wang, zwhong714, wangrui12}@sjtu.edu.cn

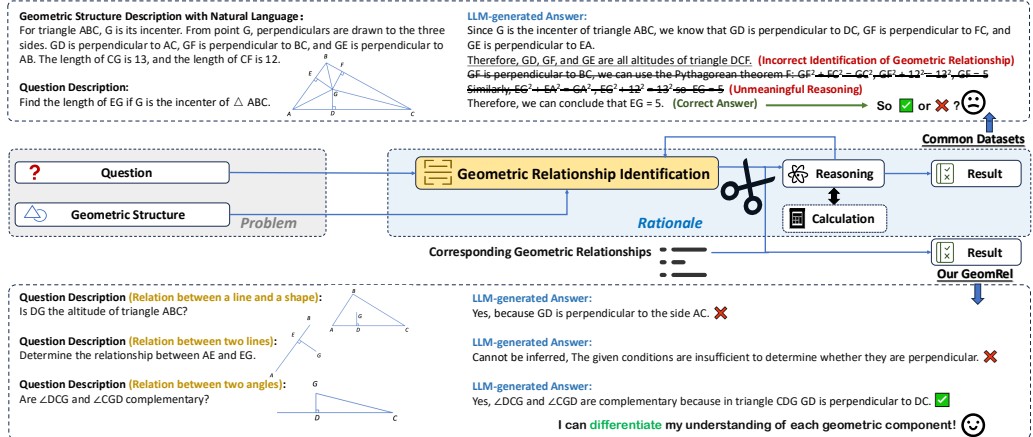

Figure 1: The general process of solving geometric problems (**Middle**). Within this process, identifying the geometric relationships is a fundamental step, one must first accurately identify the geometric structures, then apply theorems for reasoning or calculation to reach the final answer. Compared to traditional geometric datasets that only assess the accuracy of final answers (**Top**), we extract the fundamental steps of geometric relationship identification (GRI) to create the GeomRel benchmark, which evaluates whether LLMs truly understand geometric structures (**Bottom**).

## Abstract

Geometric ability is a significant challenge for large language models (LLMs) due to the need for advanced spatial comprehension and abstract thinking. Existing datasets primarily evaluate LLMs on their final answers, but they cannot truly measure their true understanding of geometric structures, as LLMs can arrive at correct answers by coincidence. To fill this gap, we introduce the GeomRel dataset, designed to evaluate LLMs' understanding of geometric structures by isolating the core step of geometric relationship identification in problem-solving. Using this benchmark, we conduct thorough evaluations of diverse LLMs and identify key limitations in understanding geometric structures. We further propose the Geometry Chain-of-Thought (GeoCoT) method, which enhances LLMs' ability to identify geometric relationships, resulting in significant performance improvements. Our work is accessible at https://github.com/banyedy/GeomRel.

## 1 Introduction

Mathematical ability has gradually become a key benchmark for evaluating large language models (LLMs) (OpenAI, 2022; Peng et al., 2023; Touvron et al., 2023a), as it can measure whether a model has preliminarily developed rigorous logic and abstract thinking (Sternberg & Ben-Zeev, 2012). Geometric ability, as an important branch, demands more spatial reasoning and a profound understanding of geometric diagrams, which presents even greater challenges for models. Recently, many efforts have focused on testing the geometric ability of LLMs, widely collecting or constructing

---

*Corresponding Author

large-scale datasets to comprehensively evaluate whether the models possess strong geometric abilities (Ye et al., 2008; Trinh et al., 2024; Zhang & Moshfeghi, 2024; Zhang et al., 2024a).

Geometric ability relies on accurately understanding geometric structures (Lindquist & Shulte, 1987). When humans tackle geometric problems, they usually start by identifying relationships within the structure, then reason and calculate based on those relationships to arrive at a final answer (Clements & Battista, 1992; Buckley et al., 2019). This process is illustrated in the middle of Figure 1. In contrast, LLMs generate answers in an end-to-end mode without explicitly decoupling these steps. They produce a blended rationale that includes the final answer, making it challenging to assess the correctness of their step-by-step reasoning. Consequently, current evaluation methods primarily calculate the accuracy by exactly matching the final answers with reference answers.

However, this risks misrepresenting the LLMs' true geometric abilities. We manually decouple the part of the LLM-generated rationale involving geometric relationship for each sample on three popular datasets, and calculate the individual accuracy of this and the final answer, respectively, as shown in Table 1. Normally, the accuracy of identifying geometric relationships should be higher than the final answer, as both geometric relationship identifications and reasoning calculations influence the latter.

Table 1: Accuracy comparisons between the final answer and GRI under three existing datasets with the GPT-3.5-Turbo model. We randomly sample 50 text-only geometry problems from each dataset, and the GRI accuracy is manually verified.

| Dataset | Answer Acc | GRI Acc |
|---|---|---|
| MATH(Hendrycks et al., 2021b) | 0.18 | 0.22 |
| Geometry3K(Lu et al., 2021) | 0.36 | 0.12 |
| PGPS9K(Zhang et al., 2023) | 0.32 | 0.14 |

However, we are surprised to find that *on these datasets, the accuracy of geometric relationship identification is significantly lower than the answer accuracy*. This indicates that in many cases, LLMs coincidentally produce correct answers despite errors in identifying geometric relationships, as one example illustrated at the top of Figure 1. These findings suggest that, under current datasets with the evaluation paradigm of exactly matching final answers, we are unable to measure whether LLMs truly understand geometric structures accurately.

To this end, we extract the sub-step of geometric relationship identification (GRI) from mainstream geometric problems and construct a dataset called GeomRel. It can serve as a minimal module for evaluating a model's ability to understand geometric structures. Although it is unable to measure more complex reasoning abilities, the measurement of geometric ability is lossless as it only involves one skill of identifying relationships, which is the foundation of reasoning ability. In Section 2, We present a detailed overview of our GeomRel dataset, including its construction, categorization, and the methods used to generate both **basic** and **advanced** versions. Our dataset forms the benchmark for testing their ability to identify and understand geometric structures.

Based on this benchmark, we comprehensively evaluate whether current LLMs understand geometric structures in Section 3. Extensive experiments on the benchmark demonstrate that:

- Current LLMs perform well in identifying simple geometric relationships but perform poorly in identifying complex structures, especially for **Angle-based** relationships. The best-performing LLM, GPT-4o, exceeds the Random performance by **48.91%** on the basic GeomRel, but only **20.34%** on the advanced GeomRel.

- We further investigate how the diversity strategies implemented in GeomRel — such as point relabeling and the incorporation of irrelevant information — affect model performances. Our findings suggest that strategically enhancing the complexity of geometric descriptions can lead to significant performance improvements.

- We examine the effects of various prompting techniques, such as Few-Shot (Brown et al., 2020) and Chain-of-Thought (Wei et al., 2022; Kojima et al., 2022), on model performances. Our findings indicate that, despite differences in reasoning lengths compared to the original prompts, these techniques do not lead to substantial improvements in geometric identification performance. Additionally, we encounter some unexpected obstacles during the forward reasoning process.

- We also fine-tune the open-source model LLaMA3-8B-Instruct using question-answer pairs but find that this does not necessarily improve its understanding of geometric structures.

Finally, in Section 4, to further enhance LLMs' ability in geometric relationship recognition, inspired by the Chain-of-Thought (CoT) technique, we propose the **Geo**metry **C**hain-**of**-**T**hought (**GeoCoT**) to elicit LLMs to identify geometry relationships step by step. The two-stage pipeline first breaks

down geometric structures into points and lines, then precisely extracts relevant information from the breakdown and applies reverse reasoning to overcome reasoning obstacles. This method substantially increases identification accuracy, with an average improvement of **9.15%** on the basic GeomRel and **14.79%** on the advanced GeomRel in the Few-Shot setting, showing a range of improvements across various domains.

## 2 GeomRel: GEOMETRIC RELATIONSHIP IDENTIFICATION BENCHMARK

To examine whether LLMs are capable of understanding geometric structures, we propose the Geometric Relationship Identification (GeomRel) benchmark. We expect that LLMs can accurately identify explicit or implicit geometric relationships according to a given geometric structure description.

A geometric relationship is formed by two identical or different geometric elements. Therefore, we first identify the most basic geometric elements and abstract several geometric relations based on them to conduct a finite geometric relationship pool. Next, we gathered geometric scenarios that could give rise to the targeted geometric relationships, starting from definitions and extending to properties. After organizing and supplementing these scenarios, we obtained the basic dataset. Using this data, we applied rule-based operations, such as condition concatenation, to generate more complex geometric structures. Additionally, we incorporated relatively independent cases, resulting in the advanced dataset. Finally, by adding scenarios where the relationships could not be determined and performing diversity-enhancing operations, we arrived at the complete GeomRel dataset. The distribution of the dataset across the main categories is shown in Table 3. The framework construction process is shown in Figure 2, and we will introduce the details in the following sections.

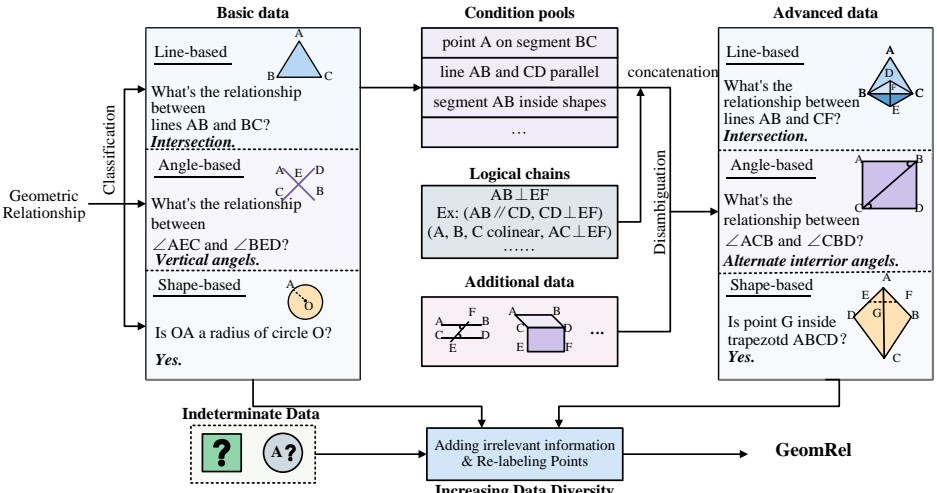

Figure 2: The GeomRel Framework

### 2.1 FINITE GEOMETRIC RELATIONSHIP POOL

Before defining the relationship pool, we first identify four fundamental geometric elements: point $p$, line $l$, angle $a$, and shape $s$, following a progressive order from zero to two dimensions. We define the spaces of point, line, angle, and shape are $\mathcal{P}$, $\mathcal{L}$, $\mathcal{A}$, and $\mathcal{S}$, respectively. They consist of the element space $\mathcal{E} = \mathcal{P} \times \mathcal{L} \times \mathcal{A} \times \mathcal{S}$.

We can theoretically combine any two types of elements (which can be the same) from $\mathcal{E}$ to create ten types of relationships ($C_4^2 + 4 = 10$). However, among these, the point-point relationship and the relationship between angles and the other three elements have no practical significance, while the shape-shape relationships may involve three-dimensional space, which we will not consider for now. Therefore, we treat the remaining five as the basic relationship types in the relationship pool, with each relationship type corresponding to several common special relationships (A total of 26 relationships), Further, we categorize them into three main domains: **Line-based**, **Angle-based**, and

**Shape-based** relationships. The hierarchical relationships and all specific relationships are shown in Table 2, which consists of our geometric relationship pool.

Table 2: The geometric relationship pool of our GeomRel dataset.

| Domain | Relationship Type | Specific Relationships |
|---|---|---|
| Line-based | Line-Point ($< \mathcal{L}, \mathcal{P} >$)
Line-Line ($< \mathcal{L}, \mathcal{L} >$) | $p$ is on $l$; $p$ is not on $l$
Parallel; Perpendicular; Intersecting |
| Angle-based | Angle-Angle ($< \mathcal{A}, \mathcal{A} >$) | Complementary; Supplementary; Corresponding; Alternate Interior; Consecutive interior; Alternate Exterior; Consecutive Exterior; Adjacent/opposite(in quadrilateral) |
| Shape-based | Shape-Point ($< \mathcal{S}, \mathcal{P} >$)
Shape-Line ($< \mathcal{S}, \mathcal{L} >$) | $p$ is inside/outside/on the boundary of $s$ (Polygon, circle)
$l$ intersects/tangent to/disjoint from $s$ (Polygon, circle); Special segment (Triangle median, Circle radius/diameter/chord, Quadrilateral side/diagonal) |

## 2.2 Basic Data Generation

After constructing the geometric relationship pool, we next generate our relationship identification data. To cover as many geometric situations that involve these relationships as possible, we used a "from definition to properties" approach (Curry, 1977). Starting from **definitions** refers to the fundamental geometric definitions, which are typically singular and relatively few in number. In contrast, deriving from **properties** involves identifying other geometric elements that exhibit characteristics leading to the same relationship, which often yields multiple cases where the relationship holds. An example are shown in Appendix C.2. This set of data serves as the foundation for our subsequent generation of more complex geometric structures, laying the groundwork for the final preparation of the basic subset in the dataset.

## 2.3 Advanced Data Generation

The advanced data is obtained through a series of operations including combination and enhancement built upon the basic data. By pooling these basic data, classifying them and combining them according to the rules we constructed, and then adding additional non-rule generated data and some diversifying operations, we obtained a complete and rich advanced dataset GeomRel-advanced (Figure 2).

**condition pools:** We organized geometric descriptions from the basic dataset that satisfy the same geometric relationship into subsets defined as a condition pool. Each data entry is structured as follows:

$$\{\text{Input} : e_{\text{in}}, \text{Condition} : c_i, \text{Output} : e_{\text{out}}\}$$

which is consolidated into a condition pool. For each pool $P_j$, all conditions within satisfy the geometric relationship $R_j$ where pair of elements ($E_{\text{in}}, E_{\text{out}}$) are involved. Additionally, since generating augmented data requires more detailed geometric information, for example, conditions that satisfy the relationship "point C is on line AB," if A and B are two different points on a

---

**Algorithm 1** Merging Geometric Conditions

1: *Input*: List of conditions *Chain*
2: *Output*: merged condition $c_{\text{merged}}$
3: $c_{\text{merged}} \leftarrow Chain[0]$
4: **for** $c$ in $Chain[1:]$ **do**
5:    consist($c, c_{merged}$)
6:    /* Modify the representation of elements in $c$ to be compatible with $c_{\text{merged}}$ */
7:    **if** $c[input] = c_{\text{merged}}[output]$ **then**
8:       Add $c[condition]$ to $c_{\text{merged}}[condition]$
9:       $c_{\text{merged}}[output] \leftarrow c[output]$
10:    **end if**
11: **end for**

---

circle O, we cannot determine the relationship between point C and circle O using the previous relationships. Therefore, we decompose, refine, and supplement some of the relationships in previous dataset. In total, 15 condition pools are created for following steps (See Appendix C.3 for examples).

**Condition concatenation:** New conditions can be concatenated in the form of conditional chains. Before we concatenate them, we need to determine the logic first. For example, if conditions in pool 1 satisfy the intersection of two lines, and conditions in pool 2 satisfy the parallelism of two lines, then by combining conditions from these two pools, we can obtain a new relationship indicating the intersection of two lines, thus combining these logics together. Another example would be if Condition 1 satisfies a line segment being inside a shape, and Condition 2 satisfies a point being

inside this line segment, then by combining them, we can conclude that a point is inside a shape. Similar logics exist, and we have created a total of 15 new logics. These logics are formed by chains of two or three geometric conditions. We demonstrate in Algorithm 1 how such conditional chains are combined to form new conditions. Specific example of our concatenation method can be found in Appendix C.4.

**Generation of Additional Data:** In addition to the data obtained through conditional concatenation, we have also generated some data that is relatively **independent** in terms of geometric structure. Particularly concerning relationships between angles, we generated simple scenarios based on examples from past datasets, and extracted the relationships therein. For instance, in scenarios where *two parallel lines intersect a third line*, we annotated the relationships between the eight angles formed by the intersection of three lines pairwise, processed them, and integrated them into the dataset.

**Disambiguation:** Due to the reliance on visual aids for past geometric relationship judgments, ambiguity may arise in purely textual geometric descriptions. We have removed or modified instances where confirmation through textual description alone was not feasible manually. For example, in cases such as the intersection of lines AB and CD, where point E lies on line CD, the relationship between point E and line AB (whether collinear) cannot be definitively established when the position of point D as an intersection is **uncertain**. Therefore, we chose to exclude such data.

## 2.4 Indeterminate Data Generation

Our geometric relationships are ultimately questioned through *multiple-choice questions*. To prevent the blind guessing of results from affecting assessments of geometric capabilities, we have included an option in each question labeled *"Cannot be inferred"*. Simultaneously, we have added some similar questions where this option is need to be used, which means the geometric relationships in this question are **not clearly defined**. By incorporating these specific question, we aim to test the model's ability to recognize such scenarios and minimize the impact of language model hallucinations.

We generate indeterminate data by removing or replacing conditions with irrelevant ones. For example, given the condition "Line AB and CD are perpendicular to line EF," we deduce that lines AB and CD are parallel. However, changing "CD perpendicular to EF" to "CD perpendicular to GH" prevents us from establishing the final relationship. Ambiguous data generated in advanced scenarios are retained using this option.

## 2.5 Increasing Data Diversity

**Adding unrelated information (UI):** After establishing the geometric information relevant to relationship determination, we enhanced the dataset by incorporating irrelevant information to make the data more realistic and comprehensive. This additional information includes irrelevant *geometric configuration information*, such as introducing new points to form new shapes or altering existing geometric relationships to create new ones. *Geometric measurement information* that does not affect the original geometric structure but adds quantitative details it also included. Adding this information enhances the diversity of the dataset and enables evaluation of LLMs' ability to filter information during assessment.

Table 3: Statistics of relation types in the dataset

| Cluster | Basic | Advaned |
|---|---|---|
| Line-based | 122 | 1168 |
| Angle-based | 103 | 275 |
| Shape-based | 108 | 853 |

**Re-labeling Points (RP):** In the previous steps, most of the data we obtained used conventional alphabetical notation to represent geometric elements, such as quadrilateral ABCD and triangle EFG. To better assess the model's generalization ability, we randomly selected and shuffled the notation of these points. For example, the notation of quadrilateral ABCD could be shuffled to quadrilateral DACB, and triangle EFG could be re-labeled as triangle EWG. This process does not alter the geometric structure but introduces variability in the notation.

Specific example can be found in Appendix C.5. By randomly sampling a proportion of the original data and applying these two operations, we expanded the original dataset by 25%, the final Statistics of the dataset are shown in Table 3.

## 3 COMPREHENSIVE EVALUATION

In this section, we comprehensively evaluate *whether LLMs can truly understand geometric structures* through their GRI abilities based on our GeomRel dataset.

### 3.1 EXPERIMENTAL SETUP

**Model and Implementation.** We comprehensively evaluate nine LLMs, encompassing both API-based models and open-source models. The API-based models include the GPT series (GPT-3.5-Turbo, GPT-4-Turbo, and GPT-4o) (Wu et al., 2023; OpenAI, 2023), the Qwen series (Qwen1.5-110B and QwenMax) (Bai et al., 2023), and Claude-3-Opus (Anthropic, 2024). The open-source models include the Llama series (LlaMA-2-13B-Chat, LlaMA-3-8B-Instruct, LlaMA-3-70B-Instruct) (Touvron et al., 2023b;a) are evaluated. For all baselines, we set temperature $\tau = 0$. We also provide random baseline comparison. The human baseline was derived from benchmark tasks completed by five science and engineering graduate students. Following this, we fine-tuned the LlaMA-3-8B-Instruct model and conducted a detailed study on its performance post fine-tuning. **Metrics.** We evaluate LLMs using large-scale comparisons between model-generated answers and standard reference answers. By modifying concise prompts, we achieve extraction accuracies exceeding 99% across various models. Our assessment extends to evaluating LLMs' accuracy across diverse domains within the dataset. Additionally, we investigate their performance in identifying cases where conditions are insufficient. Specifically, we compute precision, recall, and F1 score for the "can't be inferred" category. Precision (P) represents the *proportion of truly unidentifiable cases among those that the model failed to identify*, while Recall (R) indicates the *proportion of unidentifiable geometric structures that the model correctly flagged as unidentifiable*. The F1-score (F1) provides a balanced evaluation by tabing the the harmonic mean of P and R.

**Prompt Settings.** In large-scale model evaluation, we do not include reasoning-guiding text in the prompt (Zero-Shot prompting). While for the default large language model GPT-3.5-Turbo, we also employed other prompting approaches. Specifically, few-shot in-context learning (Few-Shot) (Brown et al., 2020), zero-shot chain-of-thought (Zero-Shot-CoT) (Kojima et al., 2023), few-shot chain-of-thought prompting (Few-Shot-CoT) (Wei et al., 2023) are leveraged to tackle various graph reasoning tasks in the GeomRel benchmark. Task instructions are simple and clear for generality, as presented in Appendix E.4.

### 3.2 MAIN RESULTS (TABLE 4)

Table 4: Accuracy performances (%) of different LLMs on our GeomRel dataset.

| Model | I. Line-based | | II. Angle-based | | III. Shape-based | | *Average* | |
|---|---|---|---|---|---|---|---|---|
| | Basic | Advanced | Basic | Advanced | Basic | Advanced | Basic | Advanced |
| GPT-4o | 77.87 | 52.91 | **66.67** | 29.00 | **87.04** | **58.38** | **77.86** | **47.93** |
| GPT-4-Turbo | **81.15** | **53.34** | **66.67** | 29.00 | 80.56 | 56.98 | 76.79 | 46.44 |
| GPT-3.5-Turbo | 65.57 | 46.23 | 54.17 | 21.75 | 72.22 | 43.26 | 63.32 | 37.08 |
| Qwen1.5-110B | 68.85 | 35.53 | 59.17 | 23.75 | 46.30 | 35.99 | 58.11 | 31.76 |
| QwenMax | 69.67 | 39.90 | 62.50 | 23.75 | 67.59 | 42.32 | 66.59 | 35.32 |
| Claude-3-Opus | 75.41 | 44.35 | 45.00 | 20.50 | 67.59 | 43.38 | 62.67 | 36.08 |
| LLaMA3-70B-Instruct | 69.67 | 38.70 | 34.17 | 27.25 | 69.44 | 40.09 | 57.76 | 35.35 |
| LLaMA3-8B-Instruct | 63.16 | 34.14 | 30.43 | **32.50** | 50.92 | 39.87 | 48.17 | 35.17 |
| LLaMA2-13B-Chat | 36.84 | 25.70 | 30.43 | 30.00 | 42.31 | 25.95 | 36.53 | 27.22 |
| Random | 28.53 | 29.43 | 25.00 | 20.00 | 33.33 | 33.33 | 28.95 | 27.59 |
| Human | 71.73 | 39.34 | 52.86 | 34.63 | 90.63 | 69.41 | 71.74 | 47.79 |

**Performances on Basic** GeomRel. Except for the earlier LLaMA2-13B-Chat model, all other models significantly surpass the Random baseline on both the Line-based and Shape-based domains. Notably, the stronger models, GPT-4o and GPT-4-Turbo, achieve over 60% accuracy across all three domains, demonstrating impressive performance. This suggests that **LLMs possess the ability to understand simple geometric structures and exhibit preliminary GRI abilities.**

**Performances on Advanced** GeomRel. We note that performances in the advanced GeomRel are significantly lower than in the basic GeomRel across all domains. Except for the GPT family of models, the results on advanced GeomRel are nearly at Random performance levels. These findings

suggest that **more complex structures pose a major challenge for accurate identification by current LLMs, highlighting their limitations in stronger GRI abilities.**

**Angle-based Relations are Particularly Difficult for LLMs.** In comparison to Line-based and Shape-based domains, LLMs demonstrate notably weaker performance in the Angle-based domain. Specifically, the performances in the advanced GeomRel approach Random levels, indicating that **Angle-based geometric relations are more difficult for LLMs to identify.**

**Evaluation Reasonableness.** As shown in Figure 3, there is a *positive correlation* between the performance of the basic and advanced datasets in line-based and shape-based tasks, indicating that an improved understanding of these foundational relations also enhances recognition of similar relations in more complex geometric structures. However, for the angle-based subset, we do not observe a positive correlation, likely due to the inability to comprehend the geometric structures present in the advanced subset.

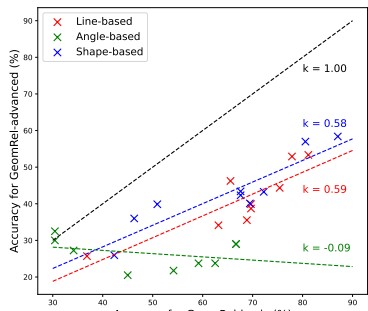

Figure 3: Accuracy correlations between basic and advanced GeomRel.

**Bias between geometric relations that LLMs subjectively fail to identify and those that are objectively unidentifiable.** In Table 5, we observe the model's performance related to unrecognizable geometric relations. Overall, the F1 performance is consistent with that on the full dataset. The stronger models, GPT-4-Turbo and GPT-4o, show a balance between precision and recall, suggesting that they can account for unidentifiable geometric structures without hastily giving a "cannot be inferred" response.

In contrast, the Qwen series and Claude-3-Opus models exhibit higher recall but lower precision, indicating a tendency to provide more uncertain answers. This may be related to additional measures these models take to handle hallucinations, leading to a lack of confidence in providing accurate answers. On the other hand, the LLaMA series models lean towards higher precision but lower recall, suggesting a tendency to force inferences even when the problem exceeds their capabilities. Examples are shown in Appendix E.3.

Table 5: LLM performances related to unrecognized geometric relations.

| Model | Basic | | | Advanced | | |
|---|---|---|---|---|---|---|
| | P | R | F1 | P | R | F1 |
| QwenMax | 0.63 | 0.78 | 0.70 | 0.32 | 0.60 | 0.42 |
| Qwen1.5-110B | 0.54 | 0.79 | 0.64 | 0.30 | **0.68** | 0.42 |
| GPT-3.5-Turbo | 0.47 | 0.43 | 0.45 | **0.63** | 0.16 | 0.25 |
| GPT-4-Turbo | 0.77 | 0.79 | 0.78 | 0.43 | 0.50 | **0.46** |
| GPT-4o | **0.79** | 0.80 | **0.79** | 0.43 | 0.48 | **0.46** |
| LLaMA3-70B | 0.43 | 0.21 | 0.29 | 0.43 | 0.14 | 0.22 |
| Claude-3-Opus | 0.65 | **0.87** | 0.74 | 0.39 | 0.56 | **0.46** |
| LLaMA3-8B | 0.30 | 0.16 | 0.21 | 0.45 | 0.14 | 0.21 |
| LLaMA2-13B | 0.67 | 0.11 | 0.18 | 0.32 | 0.05 | 0.09 |

## 3.3 ABLATION STUDY ON DATA DIVERSITY OPERATIONS

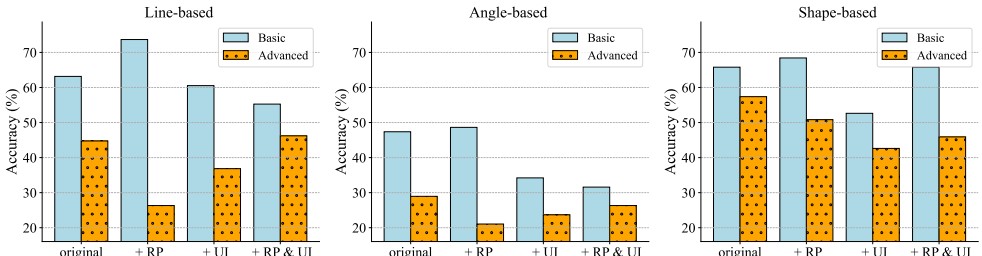

Figure 4: Ablation study about data diversity strategies of our GeomRel on GPT-3.5-Turbo model.

In Section 2.5, we design several data diversity strategies to enhance the evaluation richness of our dataset. Now, we conduct an ablation study on these strategies to explore *whether they truly have an effective differentiating effect on the LLMs' GRI abilities*.

Next, we selected a portion of the original data containing approximately 300 questions that had not undergone diversification operations, ensuring that the proportions of the subsets remained consistent with the overall dataset. This data is then subjected to three different sets of operations: the first set involved point change (RP), the second set involved adding unrelated information (UI), and the third

set applied point change after adding unrelated information (RP + UI). Along with the original data, these four sets were tested on the GPT-3.5-Turbo model.

The ablation results are shown in Figure 4. We observe that different strategies result in significant variations in the model performances, despite the core relationships in the geometric descriptions remaining consistent. Additionally, we find that **controlling the complexity of geometric descriptions leads to varying performances, which can be either positive or negative.** The same operation demonstrated inconsistent effects across different datasets. For example, applying RP to the original lines basic data improved performance, whereas it significantly decreased performance for the lines advanced data. Applying UI to the angles basic data led to a notable performance decline, but doing it to the angles advanced data with subsequent point changes resulted in some improvement.

Interestingly, although both RP and UI are designed to increase problem complexity, RP often yielded better results in over half of the comparisons. This was particularly evident for the basic data. We hypothesize that, in some cases, using more complex descriptions may stimulate LLMs' reasoning abilities, thereby enhancing performance.

## 3.4 INFLUENCE OF PROMPTING TECHNIQUES

Table 6: Accuracy performances (%) under different prompting techniques. We also report the *average token numbers* consumed during inference in parentheses. This experiment was also conducted using the GPT-3.5-Turbo model, with the same settings as before.

| Model | I. Line-based | | II. Angle-based | | III. Shape-based | |
|---|---|---|---|---|---|---|
| | Basic | Advanced | Basic | Advanced | Basic | Advanced |
| Zero-Shot | 65.57 (68) | 46.23 (156) | 54.17 (64) | 21.75 (78) | 72.22 (110) | 43.26 (180) |
| Zero-Shot-CoT | 60.71 (130) | 46.07 (277) | 46.82 (130) | 21.75 (163) | 71.48 (166) | 40.39 (311) |
| Few-Shot | 64.87 (28) | 45.75 (28) | 53.29 (31) | 24.30 (31) | 72.05 (26) | 41.52 (25) |
| Few-Shot-CoT | 70.67 (101) | 49.94 (128) | 59.20 (79) | 30.94 (134) | 71.53 (110) | 46.68 (151) |

We also analyze the prompt techniques used, aiming to explore *whether different prompt techniques would have a significant impact on the GRI abilities of LLMs*. The results are shown in Table 6.

**In-Context Learning Can be Counterproductive.** We observe that neither Zero-Shot-CoT nor Few-Shot techniques improve performance compared to Zero-Shot, with Zero-Shot-CoT even causing a significant decline in some domains. Although the *"let's think step by step"* prompting increases the length of reasoning, it appears that these additional steps do not meaningfully enhance LLMs' understanding of geometric structures and are likely ineffective in influencing results.

**Manually-guided Retionales Improve Limited Performances.** When we replace the answers in the Few-Shot examples with human-written reasoning steps, we observe some performance improvements (Few-Shot → Few-Shot-CoT), especially in the Line-based and Angle-based domains. However, this enhancement is limited, remaining within a narrow range. In the shape-based domain of the basic GeomRel, we even see a slight performance decline. These findings suggest that **while decomposing the rationale into steps can enhance performance, the gains are limited.**

**Longer Reasoning Length Does Not Mean Higher Accuracy.** In the Few-Shot setting, examples consist of straightforward question-answer pairs without additional reasoning steps, leading to the shortest reasoning lengths. Notably, this omission does not hinder the performance of LLMs, which remains on par with the Zero-Shot performances. In contrast, the other three settings incorporate reasoning steps. The Few-Shot-CoT setting exhibits a reasoning length comparable to that of the Zero-Shot setting but demonstrates better performance. In contrast, the Zero-Shot-CoT setting, despite having the

Table 7: Reasoning obstacle (in red) in the response and the correct choice (in green)

| Question | Triangle ABC is right-angled at point B. What is the relationship between line AB and line BC? |
|---|---|
| Answer choices | A. Parallel B. Perpendicular C. Intersecting but not perpendicular D. Cannot be inferred |
| Model Response | Step 1: Triangle ABC is right-angled at B, which means that angle B is a right angle. Step 2: In a right-angled triangle, the side opposite the right angle is called the hypotenuse, and the other two sides are called the legs. **Step 3: Line AB is one of the legs of triangle ABC, and line BC is the hypotenuse.** Step 4: Therefore, the relationship between line AB and line BC is that they are intersecting but not perpendicular. The answer is C. Intersecting but not perpendicular. |

longest reasoning length, yields the poorest performance. These observations indicate that **reasoning length does not directly correlate with accuracy; rather, the reasoning effectiveness has a more significant impact on performance than the mere reasoning length**.

**Reasoning obstacles exist in the forward reasoning of geometric structures.** We analyzed the reasoning processes in the Few-Shot-CoT group's results and found that reasoning obstacles appeared in all cases where the model failed to solve the problems. In these cases, what seemed to be a straightforward conclusion was unreachable for the model. As shown in Table 7, the model was unable to distinguish whether AB and BC were the legs or the hypotenuse of a right triangle, even though it previously stated that *"the side opposite the right angle is called the hypotenuse."* This outcome further suggests that large models may lack a conceptual understanding of geometric structures. See more explanations and examples in Appendix F.

## 3.5 SUPERVISED FINE-TUNING ON GeomRel

To further explore how LLMs can possess stronger GRI abilities, we try to fine-tune LLMs with our GeomRel. We split it into training, validation, and test sets in a ratio of 6:2:2, and fine-tune the Llama3-8B-Instruct model.

Figure 5 shows the Zero-Shot results, we observe that the fine-tuned Llama3-8B-Instruct does not show significant improvement over its original version and even performs worse. In the advanced GeomRel, the fine-tuned performance is almost identical to that of the original model, indicating that the fine-tuned model with explicit relationship data does not enhance their geometric reasoning capabilities. However, in the basic GeomRel, the results exhibit considerable vari-

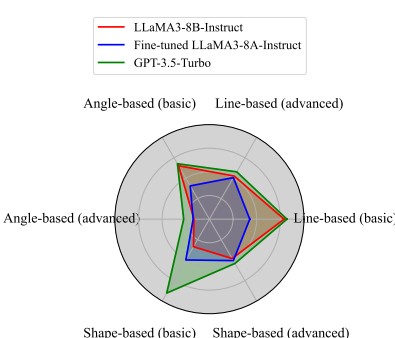

Figure 5: Comparison of Models

ability compared to the original model, with improvements in shape data but substantial declines in angles and lines data. This could be attributed to the relatively small size of the basic datasets, causing the test results to exhibit some degree of randomness after splitting. Our preliminary analysis of this data suggests that LLMs do not gain an increased understanding of geometric structures from the provided data, and fine-tuning models on QA-formed data makes it difficult to enhance their GRI abilities. We analyzed the reasons for the failure, and in Appendix E.5, we provided the fine-tuning results of other models for comparison.

## 4 GEOMETRIC CHAIN-OF-THOUGHT (GEOCOT) METHOD

In Section 3.4, we have found that the impact of using different prompting techniques on model performance was quite limited, and the conventional CoT technique failed to be effective in identifying geometric relations. To explore ways to improve the GRI abilities of LLMs, we propose a novel prompt technique for geometric problems based on CoT (GeoCoT). The pipeline and example have been illustrated in Figure 6.

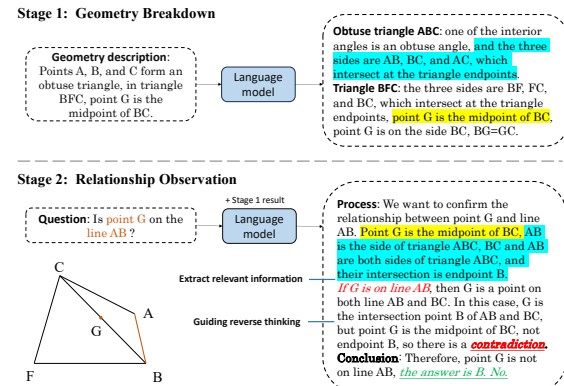

Figure 6: The pipeline and example of GeoCoT.

### 4.1 TWO-STAGE PIPELINE

We first instruct the LLMs to break down the geometric structures, decomposing the geometric information into points and lines. Then, we extract the relevant parts from the decomposed information and guide reverse reasoning based on the given question. Pipeline details are as follows.

**Stage 1: Geometry breakdown.** In the first stage, we expand the geometric information provided by the textual description. For instance, given a rectangle ABCD, we derive and list secondary conditions such as: *AB = CD, BC = AD, AB is perpendicular to BC, BC is perpendicular to CD*, and so on. This expansion ensures that the model does not overlook crucial information, thereby reducing the risk of incorrect reasoning. Our previous studies indicate that large models often fail to accurately draw

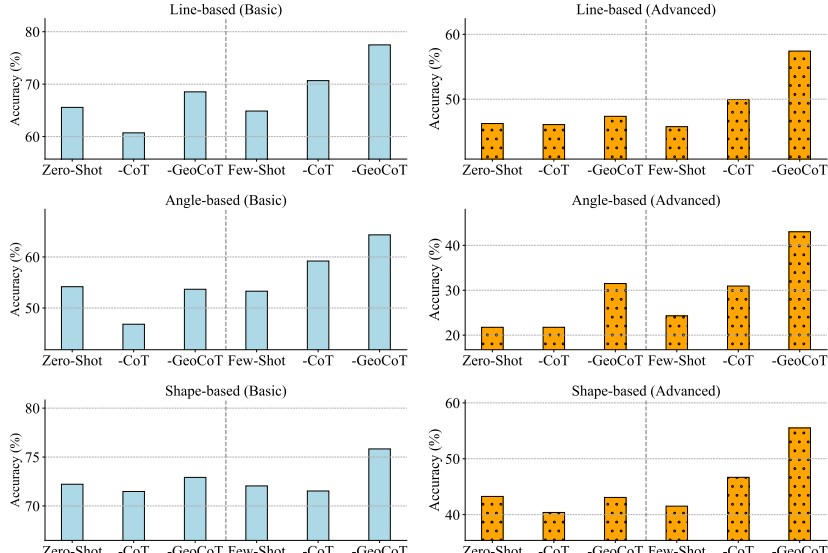

Figure 7: Model performances under our GeoCoT and other paradigms (In each subplot, the groups to the left of the dashed line use Zero-Shot, while the groups to the right use Few-Shot).

conclusions based on the provided geometric content. Therefore, by methodically listing potential secondary conditions, we enable the model to consider all relevant information during reasoning.

**Stage 2: Relationship observation.** In this stage, we re-input the result of stage 1 into the LLMs along with the problem. First, relevant information is extracted from the known conditions based on the elements mentioned in the problem. Then, reasoning is performed. To address the obstacles, we guide the model to adopt a **reverse thinking (RT)** approach. In this approach, the model assumes that a certain geometric relationship holds and then works backward, arriving at either a consistent or contradictory result, which ultimately informs the final decision.

## 4.2 RESULTS

We utilize in-context learning and create tailored examples for each domain of GeomRel following the Geo-CoT pipeline. We also develop the Zero-Shot-GeoCoT prompt and conduct experiments using the GPT-3.5-Turbo. As shown in Figure 7, GeoCoT exhibits strong performances in both Zero-Shot and Few-Shot settings, achieving an average improvement of **9.15%** on the basic GeomRel and **14.79%** on the advanced GeomRel under the Few-Shot setting. This reflects significant enhancements with our GeoCoT across various domains.

Table 8: Component ablation of GeoCOT.

| Setting | Accuracy (%) | |
|---|---|---|
| | **Basic** | **Advanced** |
| Zero-Shot-GeoCoT | 65.04 | 40.64 |
| w/o Stage 1 | 62.50 | 37.93 |
| w/o RT in Stage 2 | 64.37 | 38.81 |
| Few-Shot-GeoCoT | 72.55 | 51.98 |
| w/o Stage 1 | 63.96 | 44.58 |
| w/o RT in Stage 2 | 68.03 | 42.79 |

We further investigate the specific impact of each component in our two-stage pipeline by removing the backward reasoning guidance in the reasoning parts of Stage 1 and Stage 2 and then test their performance. The results are shown in Table 8, indicating that removing any component leads to a decline in performance. In the Zero-Shot setting, removing Stage 1 even causes the performance to be nearly indistinguishable from the original no-prompt paradigm. Some mechanisms and examples of the approach are shown in Appendix F.

## 5 CONCLUSION

In this paper, we introduced the GeomRel dataset to evaluate LLMs' geometric abilities by focusing on geometric relationship identification. Our findings highlight significant limitations in LLMs' understanding of complex geometric structures, particularly angle-based relationships. While data augmentation and prompting techniques offered limited improvements, our GeoCoT method significantly enhanced performance, improving accuracy in identifying geometric relationships. These findings underline the importance of focusing on geometric structure comprehension and offer new insights for improving LLMs' performance in this domain.

ACKNOWLEDGEMENTS

This paper is supported by the General Program of National Natural Science Foundation of China (62176153) and Ant Group.

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

## A  RELATED WORK

### A.1  EVALUATION OF LLMS ON GEOMETRY PROBLEM SOLVING

Geometric problems represent a significant and challenging aspect of mathematical problems. Several benchmarks have been developed to assess large language models' abilities in geometric problem-solving tasks, including GEOS (Seo et al., 2015), GeoShader (Alvin et al., 2017), Geometry3K (Lu et al., 2021), GeoQA (Chen et al., 2021a), GeoQA+ (Cao & Xiao, 2022), UniGeo (Chen et al., 2022), PGPS9K (Zhang et al., 2023), GeomVerse (Kazemi et al., 2023) and GeoEval(Zhang et al., 2024a). However, their evaluation primarily focuses on assessing the solutions to problems, without truly addressing the model's ability to comprehend geometric structures. The MATHVERSE (Zhang et al., 2024b) has addressed some of these issues by paying attention to the reasoning process but remains insufficient in evaluating large models' deeper understanding of geometric structures, as it lacks task decomposition. Our research begins with the fundamental elements of planar geometry and all pairwise relationships between them, constructing a specialized dataset for geometric relation recognition to specifically investigate large language models' understanding of geometric structures.

### A.2  CHAIN-OF-THOUGHT PROMPTING FOR LLMS

Recently, intriguing chain-of-thought techniques have greatly improved both the reasoning performance and interpretability of LLMs by decomposing multi-step problems into intermediate steps (Wei et al., 2022; Kojima et al., 2022; Zhang et al., 2022; Wang et al., 2022; Shi et al., 2022; Zhou et al., 2022; Lyu et al., 2023).These strategies have been widely applied to various mathematical problems. However, as a distinct type of mathematical problem, effectively understanding geometric structures is crucial. Yet, no prior work has specifically explored CoT prompting for geometric scenarios. To the best of our knowledge, we are the first to study chain-of-thought prompting for specific geometric problems, guiding LLMs to generate effective reasoning from geometric descriptions to infer geometric relations.

## B  DISCUSSIONS AND LIMITATIONS OF THE BENCHMARK

### B.1  THE INFLUENCE OF REFERENCE DIAGRAM IN GEOMETRIC REASONING

Researchers have conducted studies on the performance of LLMs in solving geometric problems that involve visual information (Zhang et al., 2024b; Wang et al., 2024), and finds that the diagrams show some differentiated enhancement effects. However, as shown in Figure 8, the presence of images makes it difficult to distinguish whether the model's success is due to observational skills or spatial imagination. Furthermore, we aim to evaluate the recognition ability of general language models, rather than focusing solely on multimodal models. In the early stages of dataset construction, we created an experimental dataset based on the Geometry3k dataset to inquire about geometric relations, using the GPT-4V model to assess differences in performance across various forms of geometric descriptions. As shown in Figure 9, the group using original diagrams (with *D (Raw)* ) performed worse than the groups using other textual geometric descriptions.

Additionally, accurate diagram-based data can only be obtained through collection, making it difficult to scale, whereas textual data can be more easily expanded. Thus, we ultimately opted for text-based geometric descriptions from a rule-based construction perspective.

**Description**: Line AB and line CD intersects at point E.
**Question**: What's the relation between ∠AEC and ∠BED?

**With diagram:**                    **Without diagram:**

"Obviously, I can **see** in the diagram that they are *vertical angles*."

"Let me **image** the diagram,
oh, I get the result, they are *vertical angles*."

Figure 8: Diagrams vs. Text: Direct observation vs. imagined visualization.

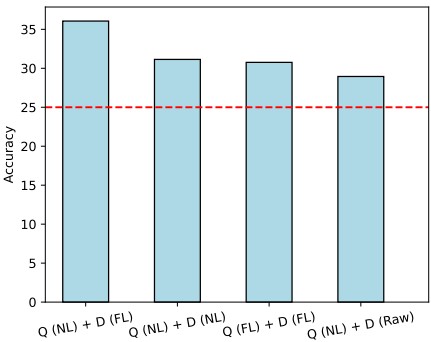

Figure 9: The accuracy of geometric relation recognition on the mini test dataset. Q represents the question, D represents the diagram, NL refers to **natural language** input, FL indicates **formal language**, and Raw represents the **original image**. The diagram data in this section is derived from the Geometry3k dataset.

### B.2 LIMITED SCOPE OF COLLECTED GEOMETRIC RELATIONS

Since our study is limited to planar geometry, while the definition of geometric problems includes solid geometry, we have not addressed this aspect due to time constraints and the greater complexity of the elements and relations involved in solid geometry. Expanding the geometric construction methods would require significant additional effort. Therefore, the problems in our dataset are restricted to *"within a plane."* Future work could further explore LLMs' understanding of three-dimensional geometric structures.

## C   DETAILS OF THE DATASET

### C.1   GEOMETRIC RELATIONSHIP CATEGORIES

**Line-based Geometric Relationships.** Line-based geometric relationships involve only the two most fundamental elements. When it comes to relationships involving points, they are often represented using points on a line, such as in the problem of collinear points. Therefore, in our dataset, we have placed these relationships into a subset. Given that these involve two basic elements, we have used enumeration to collect all geometric relationships in plane geometry that include only points and lines. This includes relationships between points, between points and lines, and between lines. Specifically, we have recorded 5 sets of relationships: three points being collinear, a point being (or not being) on a line, parallelism, intersection, and perpendicularity.

**Angle-based Geometric Relationships.** Angle-based geometric relationships require a stronger spatial understanding, as angles are formed by the intersection of lines. We primarily study the relationships between angles, including the relationships between the four angles formed by the intersection of two lines and the more complex relationships between the eight angles formed when two lines intersect with a third line. Additionally, our dataset includes specific angle relationships in particular contexts, such as adjacent angles in polygons. Overall, we have recorded 9 sets of relationships: Complementary, Supplementary, Corresponding, Alternate Interior, Consecutive Interior, Alternate Exterior, Consecutive Exterior and Adjacent/opposite angles in quadrilateral.

**Shape-based Geometric Relationships.** Shape-Based Geometric Relationships are based on figures and aim to explore the relationships between points and lines (or segments) and geometric figures. The relationships between figures and points primarily involve positional relationships, such as whether the point is inside, outside, or on the boundary of the figure. When related to lines, in addition to intersections, tangency, and disjointedness, this category also includes relationships involving special line segments within figures, such as the medians of triangles or the radii of circles. This section includes a total of 12 geometric relationships.

## C.2    BASIC DATA GENERATION

Table 9: From Definition to Properties Approach for Establishing Parallelism

| Source | Description |
|---|---|
| Definitions | "In a plane, lines AB and CD never intersect,"
"In a rectangular coordinate system, the slopes of lines AB and CD are equal." |
| Properties | "The direction vectors of lines AB and CD are proportional,"
"Quadrilateral ABCD is a parallelogram,"
"Lines AB and CD are both parallel to line EF." |

As illustrated in Figure 9, we obtain various geometric descriptions that establish the relationship "AB is parallel to CD." Starting from the definition, we obtain two distinct definitions of parallel in different situation; from the properties, we identify several fundamental cases in which parallelism can be established.

## C.3    EXAMPLES OF CONDITION POOLS

In Table 10, we present two examples of the condition pool, each containing multiple geometric descriptions.

Table 10: Examples of condition pool

| Input | Condition | Output |
|---|---|---|
| *A line segment (output) inside a shape (input)* | | |
| Triangle ABC | If ABC is a triangle, D is the midpoint of side AC | BD |
| Triangle ABC | If ABC is a triangle, M is the midpoint of side AB | CM |
| Circle O | If O is a circle with center O and radius r, and A is any point on the circumference of the circle | AO |
| Square ABCD | If ABCD is a square, E is the midpoint of side BC | AE |
| Rectangle PQRS | If PQRS is a rectangle, M is the midpoint of side PQ | MS |
| Trapezoid ABCD | If ABCD is a trapezoid with AB parallel to DC, and M is any point on side AB | CM |
| *A line (input) intersects another line (output)* | | |
| AB | Line AB intersects line CD at point E | CD |
| AB | Quadrilateral ABCD with diagonals intersecting at point E | AC |
| AB | Angle ABC is acute | BC |
| AB | Points A, B, and C form an obtuse triangle | BC |

## C.4 CONCATENATION OPERATION

As shown in Figure 11, based on the theorem that *if one line intersects another line, then its parallel line also intersects that line*, we have integrated two conditions to ultimately arrive at a relatively more complex geometric structure that satisfies the condition of *"two intersecting lines"*.

Table 11: Example of geometric concatenation operation

**Data 1** (from pool: *two parallel lines*):
Input: AB
Condition: Quadrilateral ABCD is a parallelogram
Output: CD

**Data 2** (from pool: *two intersecting lines*):
Input: AB
Condition: In triangle ABC, point D is the midpoint of BC
Output: BC

**Concatenated Data** (*two intersecting lines*):
Input: CD
Condition: Quadrilateral ABCD is a parallelogram, and in triangle ABE, point F is the midpoint of BE
Output: BE

## C.5 DIVERSIFICATION OPERATION OF THE DATASET

Table 12: Diversification of dataset examples

| Data Type | Question |
|---|---|
| **Original Data** | A circle with center A intersects AC at points E and F, triangle AGE, with GD as the altitude from vertex G to side AE. Are lines AC and AD the same line? Answer choices: A. Yes B. No C. Cannot be inferred |
| **UI** | A circle with center A intersects AC at points E and F, triangle AGE, with GD as the altitude from vertex G to side AE, and triangle AFG is equilateral, AF = 7. Are lines AC and AD the same line? Answer choices: A. Yes B. No C. Cannot be inferred |
| **RP** | A circle with center K intersects KN at points Z and T, triangle KLZ, with LP as the altitude from vertex L to side KZ. Are lines KN and KP the same line? Answer choices: A. Yes B. No C. Cannot be inferred |
| **UI + RP** | A circle with center K intersects KN at points Z and T, triangle KLZ, with LP as the altitude from vertex L to side KZ, and triangle KTL is equilateral, KT = 7. Are lines KN and KP the same line? Answer choices: A. Yes B. No C. Cannot be inferred |

As shown in Figure 12, we increased the complexity of the geometric descriptions and enhanced the diversity of the dataset through Adding unrelated information (UI) and Re-labeling Points (RP).

## D CORRELATION WITH COMMON DATASETS

In Figure 10, we compare the performance of various models on our dataset (including both the basic and advanced subsets) with several widely-used benchmark datasets. We observe a high degree of correlation in performance, particularly with math-related benchmarks, further validating the effectiveness of our dataset.

## E EVALUATION DETAILS

### E.1 MODEL HYPERPARAMETERS

Table 13 presents the complete list of hyperparameters applied to the models (including the hyperparameters of the fine-tuning operations) throughout the evaluation phase.

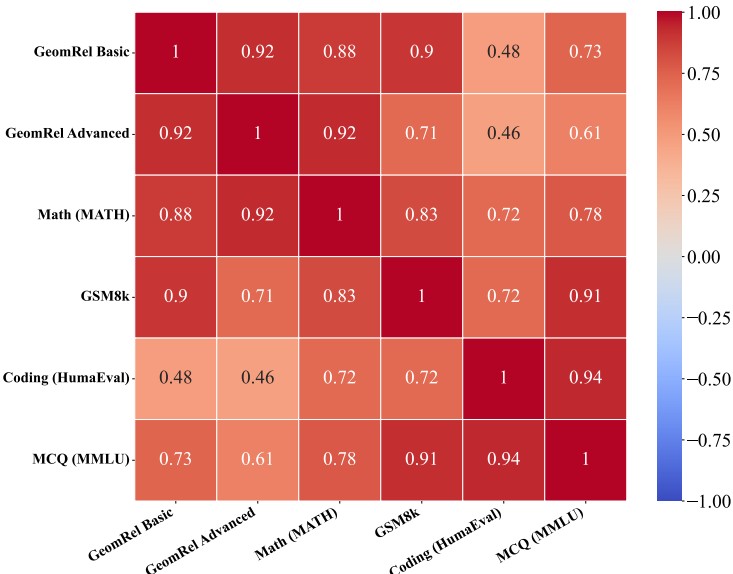

Figure 10: Correlation heatmap of model performances. The benchmarks we compared are MATH Hendrycks et al. (2021b), GSM8K (Cobbe et al., 2021), HumanEval (Hendrycks et al., 2021a) and MMLU (Chen et al., 2021b). The models used and experiment settings are consistent with those described in Section 3.1.

Table 13: Hyperparameters of Each Model

| Model Name | Parameters | Comments |
|---|---|---|
| GPT-4o | "temperature": 0, "max_tokens": 1024 | version = "gpt-4o-2024-05-13" |
| GPT-4-turbo | "temperature": 0, "max_tokens": 1024 | version = "GPT-4-turbo" |
| GPT-3.5-turbo | "temperature": 0, "max_tokens": 1024 | version = "gpt-3.5-turbo-0125" |
| Qwen1.5-110B | "temperature": 0, "max_tokens": 1024 | version = "qwen1.5-110b-chat" |
| QwenMax | "temperature": 0, "max_tokens": 1024 | version = "qwen-max" |
| Claude-3-Opus | "temperature": 0, "max_tokens": 1024 | version = "claude-3-opus-20240229" |
| LLaMA2-13B-Chat | "temperature": 0, "max_tokens": 1024 | model = "Llama-2-13b-chat" |
| LLaMA3-70B-Instruct | "temperature": 0, "max_tokens": 1024 | model = "Llama-3-70B-Instruct" |
| LLaMA3-8B-Instruct | "temperature": 0, "max_tokens": 1024 | model = "Llama-3-8B-Instruct" |
| Qwen2-7B-Instruct | "temperature": 0, "max_tokens": 1024 | model = "Qwen2-7B-Instruct" |
| LLaMA3-8B-Base-FT | "temperature": 0, "max_tokens": 1024, train_batch_size: 4,"finetuning_type": lora, "learning_rate": 1.0e-4, "num_train_epochs": 10.0, "bf16": true | model = "Llama-3-8B" |
| LLaMA3-8B-Instruct-FT | "temperature": 0, "max_tokens": 1024,"train_batch_size": 4,"finetuning_type": lora, "learning_rate": 1.0e-4, "num_train_epochs": 10.0, "bf16": true | model = "Llama-3-8B-Instruct" |
| Qwen2-7B-Instruct-FT | "temperature": 0, "max_tokens": 1024,"train_batch_size": 4,"finetuning_type": lora, "learning_rate": 1.0e-4, "num_train_epochs": 10.0, "bf16": true | model = "Qwen2-7B-Instruct" |
| Qwen2-7B-Base-FT | "temperature": 0, "max_tokens": 1024,"train_batch_size": 4,"finetuning_type": lora, "learning_rate": 1.0e-4, "num_train_epochs": 10.0, "bf16": true | model = "Qwen2-7B" |

### E.2    ZERO-SHOT RESULT OF DIFFERENT MODELS

Tables 17–21 present examples from different subsets under the Zero-Shot settings of 3 LLMs, we can observe the performance differences of models across different subsets. In Table 22, we provide multiple examples from the advanced shape-based subset and summarize the responses of three models, where we can observe that GPT-4o demonstrates significantly superior performance.

We conducted additional experiments on the **GPT-3.5-turbo** model using multiple sampling runs (temperature set to 1). Its performance was evaluated using *pass@k* and self-consistency with majority voting over $k$ generations (denoted as *acc(SC=k)*). The results are summarized in Table 14.

We observed that *pass@k* increases rapidly with larger $k$, indicating a growing likelihood of generating the correct answer through multiple attempts. However, the majority voting accuracy (*acc(SC=k)*) does not improve significantly, suggesting that the model's probability of generating correct answers remains low. This indicates that its reasoning on geometric tasks is inconsistent, leading to dispersed outputs.

Table 14: Results(%) of GPT-3.5-turbo using multiple sampling runs

| k | I. Line-based | | II. Angle-based | | III. Shape-based | |
|---|---|---|---|---|---|---|
| (pass@k \| acc(SC=k)) | Basic | Advanced | Basic | Advanced | Basic | Advanced |
| 1 | 64.29 \| 64.29 | 42.46 \| 42.46 | 42.11 \| 42.11 | 22.41 \| 22.41 | 68.75 \| 68.75 | 38.31 \| 38.31 |
| 3 | 71.43 \| 60.71 | 62.30 \| 45.24 | 63.16 \| 52.63 | 37.93 \| 22.41 | 93.75 \| 75.00 | 59.74 \| 38.96 |
| 5 | 75.00 \| 64.29 | 72.22 \| 44.44 | 68.42 \| 42.11 | 55.17 \| 25.86 | 100.00 \| 75.00 | 68.83 \| 43.51 |
| 10 | 75.00 \| 64.29 | 83.33 \| 45.24 | 84.21 \| 57.89 | 70.69 \| 32.76 | 100.00 \| 75.00 | 77.92 \| 42.21 |

## E.3 OBJECTIVELY UNIDENTIFIABLE OR SUBJECTIVELY UNIDENTIFIABLE

Table 23 presents the QwenMax's Zero-Shot responses, with the four examples corresponding to: unidentifiable relation with a definite model conclusion, unidentifiable relation with no definite model conclusion, identifiable relation with no definite model conclusion, and identifiable relation with a definite model conclusion. We observe that the model may incorrectly judge a geometric structure as incomplete, or it may make an arbitrary conclusion about a geometric structure where information is indeed insufficient.

## E.4 DIFFERENT PROMPTS STRATEGY'S RESULTS ON GPT-3.5-TURBO.

In Table 25, we can observe the differences in response lengths across various prompting strategies. In this example, the Zero-Shot group is the only one that produced a correct answer, while all other strategies resulted in incorrect answers. This indicates that prompting strategies can sometimes lead to poorer outcomes.

## E.5 ADDITIONAL FINE-TUNING RESULTS

In response to further experiments and suggestions, we present updated fine-tuning results for various models on the GeomRel dataset. The results are shown in Table 15.

Table 15: Fine-tuning Results for Different Models on GeomRel

| Model | I. Line-based | | II. Angle-based | | III. Shape-based | |
|---|---|---|---|---|---|---|
| | Basic | Advanced | Basic | Advanced | Basic | Advanced |
| LLaMA3-8B-Instruct | 63.16 | 42.11 | 52.17 | 13.04 | 26.92 | 38.46 |
| LLaMA3-8B-Instruct-FT | 34.14 | 40.56 | 32.50 | 13.75 | 39.87 | 40.51 |
| LLaMA3-8B-Base-FT | 86.86 | **71.67** | 63.16 | **54.90** | 93.75 | 61.69 |
| Qwen2-7B-Instruct | 65.55 | 31.84 | 53.75 | 20.71 | 55.38 | 34.69 |
| Qwen2-7B-Instruct-FT | 82.14 | 70.87 | **78.95** | 54.44 | 94.48 | 68.83 |
| Qwen2-7B-Base-FT | **89.29** | 70.48 | 73.68 | 51.51 | **95.70** | **70.13** |
| GPT-4o | 77.87 | 52.91 | 66.67 | 29.00 | 87.04 | 53.38 |

- **Performance of Fine-tuning:** Fine-tuning on **LLaMA3-8B-Base** significantly improves performance across all subsets, outperforming **GPT-4o** in most categories. However, fine-tuning on **LLaMA3-8B-Instruct** results in lower performance, likely due to the model's tendency to select uncertain responses, such as "cannot be inferred," when faced with difficult cases.

- **Comparison of Qwen2-7B Models:** Both **Qwen2-7B-Base** and **Qwen2-7B-Instruct** demonstrate strong performance after fine-tuning, with the Base model showing slightly better results in most cases. This contrasts with LLaMA models, where the fine-tuned Base model consistently outperforms the fine-tuned Instruct model. The difference may be due to variations in instruction-tuning methodologies and content.

- **Importance of Model Selection:** The results highlight the importance of carefully selecting models for fine-tuning on GeomRel. While LLaMA models show substantial improvements after fine-tuning, Qwen models exhibit comparable or superior performance in several categories.

These findings emphasize the importance of both model selection and fine-tuning strategies in achieving optimal performance on the GeomRel dataset.

We conducted further analysis on the fine-tuning results of the **LLaMA3-8B-Instruct** model and observed that its marginal gains were due to the model frequently outputting the uncertain answer, "cannot be inferred." Additionally, we fine-tuned the **LLaMA3-8B-Base** model, which showed substantial improvement, achieving 81.26% and 62.75% accuracy on the Basic and Advanced subsets, respectively, surpassing **GPT-4o**.

To better understand this phenomenon, we analyzed the proportion of "cannot be inferred" responses generated by each model, as shown in Table 16.

Table 16: Proportion of "cannot be inferred" responses across models.

| Model | Basic | Advanced |
|---|---|---|
| LLaMA3-8B-Instruct | 14.71% | 8.21% |
| Fine-tuned LLaMA3-8B-Instruct | 71.53% | 67.40% |
| Fine-tuned LLaMA3-8B-Base | 25.03% | 21.41% |
| Truth | 27.94% | 26.49% |

The results reveal that the fine-tuned **LLaMA3-8B-Instruct** model heavily leaned toward uncertain answers, with a significantly higher proportion of "cannot be inferred" responses compared to the actual data distribution. In contrast, the fine-tuned **LLaMA3-8B-Base** model closely aligned with the actual proportions in the dataset.

This discrepancy highlights two critical factors:

- **The impact of fine-tuning data:** The LLaMA3-8B-Instruct model's performance suggests that its initial instruction-tuning introduced a bias toward cautious or uncertain answers.

- **The alignment of model outputs with data distributions:** Fine-tuning the LLaMA3-8B-Base model led to outputs more consistent with the actual dataset proportions, contributing to its superior performance.

These findings underscore the importance of selecting appropriate base models for fine-tuning, especially for tasks where output distribution plays a crucial role in overall accuracy.

## F  REASONING OBSTACLES AND THE EFFECTIVENESS OF OUR GEOCOT APPROACH

To better understand the forward reasoning obstacles mentioned in Section 3.4 and the effectiveness of our GeoCoT method proposed in Section 6, we attempt to analyze from the perspective of reasoning pathways.

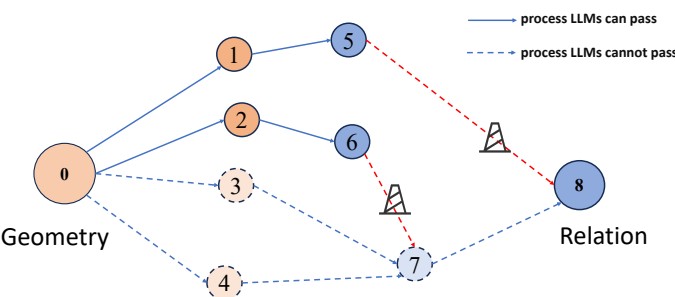

Figure 11: Forward reasoning process of LLMs

As shown in Figure 11, during the model's forward reasoning, when both the problem and the geometric description are provided, the model tends to generate a reasoning process more directly

related to the problem ($0 \to 1$, $0 \to 2$, $2 \to 6$, $1 \to 5$), thereby overlooking some implicit inferences ($0 \to 3$, $0 \to 4$). This leads to the failure in deriving conclusion 7, which in turn prevents the final answer from being reached. The model, relying solely on 5 and 6, arrives at a final result that ultimately leads to an incorrect answer. Specific examples of errors made by GPT-3.5-turbo in the Few-shot-CoT setting in Table 27.

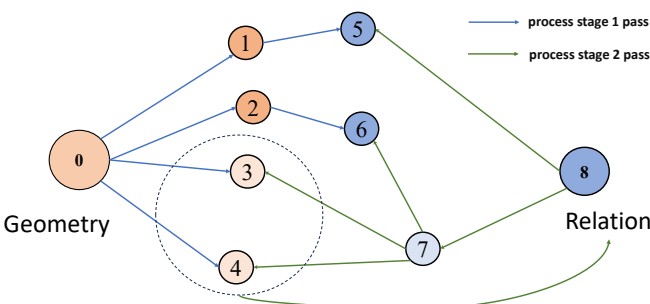

Figure 12: Reasoning process with GeoCoT

However, as shown in Figure 12, after applying GeoCoT, in stage 1, we ignore the specific problem and focus solely on the geometric description to extract as much explicit and implicit information as possible ($0 \to 1$, $0 \to 2$, $0 \to 3$, $0 \to 4$, $1 \to 5$, $2 \to 6$). In Stage 2, we adopt a reverse thinking approach by assuming the result and reasoning backward from it ($8 \to 5$, $8 \to 7$, $7 \to 6$, $7 \to 3$, $7 \to 4$). We then examine 3 and 4 for consistency or contradiction, allowing us to make a judgment on our initial assumption and derive the conclusion ($3, 4 \to 8$). This method effectively overcomes the obstacles encountered in forward reasoning. A comparative example is shown in Figure 7. Specific examples of GPT-3.5-turbo's performance under CoT and GeoCoT are shown in Table 28 and Table 29, we can observe the differences in their reasoning and the correctness of the answers.

To further support our explanations, we introduced a new metric, Necessary Condition Coverage (NCC), which measures the proportion of necessary intermediate conditions covered during the reasoning process that lead to the correct answer. We evaluated this on 24 randomly selected questions using GPT-3.5-turbo with Few-Shot CoT and Few-Shot GeoCoT methods. The results are shown in Table 26.

The results demonstrate that our two-stage GeoCoT method improves NCC, particularly for advanced problems. This highlights the effectiveness of decomposing geometry problems and incorporating reverse reasoning to enhance coverage and reasoning quality.

Table 17: Zero-shot results of various LLMs on the basic line-based subset

---

**Question:**

Quadrilateral ABCD is a rectangle. What is the relationship between line AB and line BC?

**Answer choices:**

A. Parallel

B. Perpendicular

C. Intersecting but not perpendicular

D. Cannot be inferred

**Gold Answer:**

B. Perpendicular

**GPT-3.5-Turbo:**

The relationship between line AB and line BC is A. Parallel.

My choice is A. Parallel.

✘

**GPT-4o:**

In a rectangle, all interior angles are right angles (90 degrees). This means that each pair of adjacent sides meets at a right angle.

Given that quadrilateral ABCD is a rectangle:

- Line AB and line BC are adjacent sides of the rectangle.

- Since the interior angles of a rectangle are 90 degrees, line AB and line BC must meet at a right angle.

Therefore, the relationship between line AB and line BC is that they are perpendicular to each other.

The correct answer is:

B. Perpendicular

✔

**LLaMA-3-70B-Instruct:**

Since ABCD is a rectangle, we know that it has four right angles (90-degree angles).

Let's analyze the relationship between line AB and line BC:

- AB is one side of the rectangle, and BC is an adjacent side.

- In a rectangle, adjacent sides are perpendicular to each other, which means they form a right angle (90-degree angle).

Therefore, the correct answer is:

B. Perpendicular

✔

---

**Question:**

Line AB intersects circle O at two points, point C and point D. What is the relationship between line AB and line OD?

**Answer choices:**

A. Parallel

B. Perpendicular

C. Intersecting but not perpendicular

D. Cannot be inferred

**Gold Answer:**

C. Intersecting but not perpendicular

**GPT-3.5-Turbo:**

Line AB and line OD are intersecting but not perpendicular.

Choice C. Intersecting but not perpendicular.

✔

**GPT-4o:**

To determine the relationship between line AB and line OD, let's analyze the given information:

1. Line AB intersects circle O at two points, C and D. This means that line AB is a secant line to the circle.

2. Point O is the center of the circle.

Given that line AB intersects the circle at points C and D, we can infer that line OD is a radius of the circle, where D is a point on the circumference.

For line AB to intersect the circle at two points, it must pass through the circle, making it a secant line. The radius OD is perpendicular to the tangent at point D. However, since AB is a secant line and not necessarily a tangent, we need to consider the general case.

In the general case, the relationship between a secant line (AB) and a radius (OD) that intersects the secant line at one of the points of intersection (D) is that they are perpendicular.

Therefore, the correct answer is:

B. Perpendicular

✘

**LLaMA-3-70B-Instruct:**

When a line intersects a circle at two points, it is called a secant line. In this case, line AB is a secant line that intersects circle O at points C and D.

Now, let's analyze the relationship between line AB and line OD. Since point D lies on circle O, the line joining the center of the circle (O) to point D (line OD) is a radius of the circle.

The key property here is that a radius is always perpendicular to a tangent line at the point of tangency. Since line AB is a secant line, it can be thought of as a limiting case of a tangent line, where the tangent line intersects the circle at two points instead of one.

Therefore, we can infer that line OD (the radius) is perpendicular to line AB (the secant line) at point D.

My choice is:

B. Perpendicular

✘

---

Table 18: Zero-shot results of various LLMs on the basic angle-based subset

---

**Question:**
In a plane, there is a quadrilateral ABCD, and point E is a point in the plane such that AE is perpendicular to BC and DE is perpendicular to AD. What is the relationship between $\angle BED$ and $\angle AED$?
**Answer choices:**
A. Vertical angles
B. Supplementary angles
C. None of the above
D. Cannot be inferred

---

**Gold Answer:**
B. Supplementary angles

---

**GPT-3.5-Turbo:**
Since AE is perpendicular to BC and DE is perpendicular to AD, we can see that $\angle AED$ and $\angle BED$ are both right angles. Therefore, the relationship between $\angle BED$ and $\angle AED$ is that they are vertical angles.
Choice A. Vertical angles
✘

---

**GPT-4o:**
To determine the relationship between $\angle BED$ and $\angle AED$, let's analyze the given conditions:
1. AE is perpendicular to BC.
2. DE is perpendicular to AD.
Given these conditions, we can infer that:
- $\angle AEB = 90°$
- $\angle AED = 90°$
Since both angles are right angles, they are not vertical angles and not supplementary (angles that add up to 180 degrees).
Therefore, the correct answer is:
C. None of the above
✘

---

**LLaMA-3-70B-Instruct:**
Let's analyze the situation:
- AE is perpendicular to BC, and DE is perpendicular to AD.
- $\angle BED$ and $\angle AED$ are formed by the transversals AE and DE, respectively.
Since AE and DE are perpendicular to their respective lines, $\angle BED$ and $\angle AED$ are alternate interior angles. Therefore, $\angle BED$ and $\angle AED$ are equal.
My choice is:
A. Vertical angles
✘

---

**Question:**
Quadrilateral DEFG is a rectangle with DE=2 and FG=4. What is the relationship between $\angle FGD$ and $\angle EFG$?
**Answer choices:**
A. Diagonal angles
B. Adjacent angles
C. None of the above
D. Cannot be inferred

---

**Gold Answer:**
B. Adjacent angles

---

**GPT-3.5-Turbo:**
The relationship between $\angle FGD$ and $\angle EFG$ is that they are adjacent angles.
So the correct answer is:
B. Adjacent angles
✔

---

**GPT-4o:**
To determine the relationship between $\angle FGD$ and $\angle EFG$ in the rectangle DEFG, let's analyze the properties:
- $\angle FGD$ is an angle at vertex G, formed by FG and GD.
- $\angle EFG$ is an angle at vertex F, formed by EF and FG.
Since they share the side FG and are adjacent, the correct relationship is that they are adjacent angles.
Answer choice: B. Adjacent angles
✔

---

**LLaMA-3-70B-Instruct:**
In a rectangle, all angles are right angles. Since DEFG is a rectangle:
- $\angle FGD = 90°$
- $\angle EFG = 90°$
They are not diagonal angles but are adjacent because they share the side FG and vertex F.
So, the correct answer is:
B. Adjacent angles
✔

---

Table 19: Zero-shot results of various LLMs on the advanced angle-based subset

**Question:**
Line AB intersects line CD at point E, line GH intersects line CD at point F, GH is parallel to AB. What is the relationship between ∠GFC and ∠BEC?
**Answer choices:**
A. Supplementary angles
B. Alternate exterior angles
C. Vertical angles
D. None of the above
E. Cannot be inferred

**Gold Answer:**
B. Alternate exterior angles

**GPT-3.5-Turbo:**
The relationship between ∠GFC and ∠BEC is that they are alternate exterior angles. When two parallel lines are intersected by a transversal, alternate exterior angles are congruent. Therefore, the correct answer is:
B. Alternate exterior angles
✔

**GPT-4o:**
To determine the relationship between ∠GFC and ∠BEC, let's analyze the given information step by step:
1. **Lines and Intersections:**
- Line AB intersects line CD at point E.
- Line GH intersects line CD at point F.
- Line GH is parallel to line AB.
2. **Parallel Lines and Transversals:**
- Since GH is parallel to AB, and CD acts as a transversal line intersecting both GH and AB, we can use properties of parallel lines and transversals to determine angle relationships.
3. **Angle Relationships:**
- ∠BEC is formed at the intersection of lines AB and CD.
- ∠GFC is formed at the intersection of lines GH and CD.
4. **Using Parallel Line Properties:**
- ∠BEC and ∠GFC are on opposite sides of the transversal CD and outside the parallel lines GH and AB. Therefore, they are alternate exterior angles.
Given this analysis, the relationship between ∠GFC and ∠BEC is that they are alternate exterior angles.
The correct answer is:
B. Alternate exterior angles
✔

**LLaMA-3-70B-Instruct:**
A nice geometry problem!
Let's analyze the situation:
- Line AB intersects line CD at point E, so ∠BEC is an interior angle.
- Line GH is parallel to AB, so GH // AB.
- Line GH intersects line CD at point F, so ∠GFC is also an interior angle.
Now, we can use the fact that GH // AB to conclude that the corresponding angles formed by these two lines with CD are equal. Specifically, ∠GFC and ∠BEC are corresponding angles.
Corresponding angles are equal, so we have:
∠GFC = ∠BEC
Therefore, the relationship between ∠GFC and ∠BEC is that they are equal.
My choice is: None of the above (since "equal" is not among the options).
✘

**Question:**
Quadrilateral TXOF is a rectangle, TO is one of its diagonals. What is the relationship between ∠TOF and ∠TOX?
**Answer choices:**
A. Alternate interior angles
B. Complementary angles
C. Supplementary angles
D. None of the above
E. Cannot be inferred

**Gold Answer:**
B. Complementary angles

**GPT-3.5-Turbo:**
The relationship between ∠TOF and ∠TOX is that they are complementary angles. This is because in a rectangle, the diagonals bisect each other, so ∠TOF and ∠TOX are adjacent angles that form a right angle. Therefore, the sum of their measures is 90 degrees, making them complementary angles.
Choice B. Complementary angles
✔

**GPT-4o:**
To determine the relationship between ∠TOF and ∠TOX in rectangle TXOF, let's analyze the properties of a rectangle and its diagonals.
1. **Properties of a Rectangle:**
- A rectangle has four right angles (each 90 degrees).
- Opposite sides are equal and parallel.
- The diagonals of a rectangle are equal in length and bisect each other.
2. **Diagonals in a Rectangle:**
- When diagonals intersect, they form two congruent triangles.
- Each diagonal divides the rectangle into two right triangles.
3. **Angles Formed by the Diagonal:**
- ∠TOF and ∠TOX are angles formed by the diagonal TO with the sides of the rectangle.
- Since TO is a diagonal, it bisects the rectangle into two congruent right triangles, and each angle at the intersection of the diagonal with the sides of the rectangle is part of these right triangles.
4. **Relationship Between ∠TOF and ∠TOX:**
- In a rectangle, the sum of the angles around a point is 360 degrees.
- Since TO is a diagonal, it splits the rectangle into two right triangles, and the angles around point T (or O) must sum to 180 degrees because they are supplementary angles.
Therefore, ∠TOF and ∠TOX are supplementary angles because they add up to 180 degrees.
**Answer choice: C. Supplementary angles**
✘

**LLaMA-3-70B-Instruct:**
Since TXOF is a rectangle, we know that it has two sets of opposite right angles. Let's focus on the right angles at vertices T and O.
∠TOF and ∠TOX are adjacent angles that form a straight line at vertex O. We know that the sum of the measures of adjacent angles that form a straight line is 180°.
Since ∠TOF and ∠TOX are adjacent angles that form a straight line, their sum is 180°. This means that they are supplementary angles.
My choice is: C. Supplementary angles.
✘

Table 20: Zero-shot Results of various LLMs on the basic shape-based subset

| |
| --- |
| **Question:** |
| If PIC is a triangle, point U is on the extension line of side IC, CU = $\frac{1}{2}$ * IC. Is line segment PU inside triangle PIC? |
| **Answer choices:** |
| A. Yes |
| B. No |
| C. Cannot be inferred |
| **Gold Answer:** |
| B. No |
| **GPT-3.5-Turbo:** |
| Since point U is on the extension line of side IC and CU = $\frac{1}{2}$ * IC, this means that point U is located outside of triangle PIC. Therefore, line segment PU is also outside of triangle PIC. |
| So, the answer is B. No. |
| ✔ |
| **GPT-4o:** |
| To determine whether line segment PU is inside triangle PIC, let's analyze the given information step by step. |
| 1. **Triangle PIC**: We have a triangle with vertices P, I, and C. |
| 2. **Point U on the extension of side IC**: Point U is located on the line extending from side IC such that CU = $\frac{1}{2}$ * IC. This means U is outside triangle PIC because it lies on the extension of side IC beyond point C. |
| 3. **Position of PU relative to triangle PIC**: Since U is outside the triangle, we need to consider the position of line segment PU. |
| - If U is on the extension of IC beyond C, then U is not between I and C but rather beyond C. |
| - Therefore, line segment PU will extend from point P to a point U that is outside the triangle. |
| Given that U is outside the triangle, line segment PU cannot be entirely contained within triangle PIC. It must pass outside the triangle to reach U. |
| Thus, the correct answer is: |
| B. No |
| ✔ |
| **LLaMA-3-70B-Instruct:** |
| A nice geometry problem! |
| Let's break it down step by step: |
| 1. We know that point U is on the extension line of side IC, which means that IU is a straight line. |
| 2. We also know that CU = $\frac{1}{2}$ * IC, which implies that point C is the midpoint of IU. |
| 3. Now, consider the triangle PIC. Since C is the midpoint of IU, we can draw a perpendicular bisector from C to IC, which intersects IC at its midpoint, say, M. |
| 4. Since CM is perpendicular to IC, we can conclude that CM is also perpendicular to PU (as PU is a part of IU). |
| 5. Now, observe that triangle PMC is a right triangle with a right angle at M. We know that PM < IC (since P is inside the triangle PIC), and CM = $\frac{1}{2}$ * IC (since C is the midpoint of IU). |
| 6. Using the Pythagorean theorem in triangle PMC, we get: PM² + CM² = PC². |
| 7. Since CM = $\frac{1}{2}$ * IC, we can substitute this value into the equation: PM² + $\frac{1}{4}$ * IC² = PC². |
| 8. Simplifying the equation, we get: PM² + $\frac{1}{4}$ * IC² = PC². |
| 9. Since PM < IC, we can conclude that PM² < IC². Adding $\frac{1}{4}$ * IC² to both sides, we get: PM² + $\frac{1}{4}$ * IC² < IC² + $\frac{1}{4}$ * IC². |
| 10. Simplifying the inequality, we get: PC² < $\frac{5}{4}$ * IC². |
| 11. Since PC is a side of triangle PIC, we know that PC < IC (otherwise, PC would be longer than the side IC, which is not possible in a triangle). |
| 12. Combining the results from steps 10 and 11, we can conclude that PC² < $\frac{5}{4}$ * IC² < IC², which implies that PC < IC. |
| Now, here's the crucial part: since PC < IC, we can conclude that point P lies inside the triangle PIC. |
| Therefore, my answer is: |
| A. Yes |
| ✘ |

Table 21: Zero-shot Results of various LLMs on the advanced shape-based subset

| |
|---|
| **Question:** |
| If HNFO is a rectangle, point B is on the extension line of side HO, OB = HO, triangle OMB is right-angled at O, with ME being the altitude drawn from vertex M to OB, and point E is the foot of this altitude. What is the relationship between point E and rectangle HNFO? |
| **Answer choices:** |
| A. Inside B. Outside C. Cannot be inferred |
| **Gold Answer:** |
| B. Outside |
| **GPT-3.5-Turbo:** |
| Point E is inside rectangle HNFO. |
| Choice: A. Inside |
| ✘ |
| **GPT-4o:** |
| To determine the relationship between point E and rectangle HNFO, let's analyze the given information step by step. |
| 1. **HNFO is a rectangle**: This means that all angles in the rectangle are right angles, and opposite sides are equal in length. |
| 2. **Point B is on the extension line of side HO, and OB = HO**: This implies that point B is collinear with points H and O, and B is located such that the distance OB is equal to the length of side HO of the rectangle. |
| 3. **Triangle OMB is right-angled at O**: This means that angle OMB is a right angle (90 degrees). |
| 4. **ME is the altitude drawn from vertex M to OB, and point E is the foot of this altitude**: This means that ME is perpendicular to OB, and E is the point where this perpendicular line intersects OB. |
| Since OB is an extension of side HO and OB = HO, point B lies outside the rectangle HNFO. Since triangle OMB is right-angled at O, point M must be somewhere such that OM is perpendicular to OB. The altitude ME from M to OB will intersect OB at point E. |
| Since OB is an extension of side HO and point B is outside the rectangle, the altitude ME from M to OB will also be outside the rectangle. Therefore, point E, being the foot of this altitude, must also lie outside the rectangle HNFO. Thus, the relationship between point E and rectangle HNFO is: |
| Answer choice: B. Outside |
| ✔ |
| **LLaMA-3-70B-Instruct:** |
| A nice geometry problem! |
| Let's break it down step by step: |
| 1. HNFO is a rectangle, so all its angles are right angles. |
| 2. Point B is on the extension line of side HO, and OB = HO, which means OB is a continuation of HO. |
| 3. Triangle OMB is right-angled at O, which means ∠MOB = 90°. |
| 4. ME is the altitude drawn from vertex M to OB, which means ME is perpendicular to OB. |
| 5. Point E is the foot of this altitude, which means E lies on OB. |
| Now, let's analyze the situation: |
| Since ME is perpendicular to OB, and OB is a continuation of HO, ME is also perpendicular to HO. This means ME is parallel to FN, since HO and FN are opposite sides of the rectangle HNFO. |
| As ME is parallel to FN, and E lies on OB, we can conclude that E lies on the extension of FN, which means E lies on the side NF of the rectangle HNFO. |
| Therefore, the relationship between point E and rectangle HNFO is that point E lies **on** the rectangle. |
| My answer choice is: A. Inside |
| ✘ |

Table 22: Zero-shot results of various LLMs on the advanced shape-based subset with simplified answers (The correct options are highlighted in red).

| Question | GPT-3.5-Turbo | GPT-4o | LLaMA-3-70b |
|---|---|---|---|
| If HNFO is a rectangle, point B is on the extension line of side HO, OB = HO, triangle OMB is right-angled at O, with ME being the altitude drawn from vertex M to OB, and point E is the foot of this altitude. What is the relationship between point E and rectangle HNFO? Answer choices: A. Inside B. Outside C. Cannot be inferred | Point E is inside rectangle HNFO. Choice: A. Inside ✘ | Point E, being the foot of the altitude, must also lie outside the rectangle HNFO. Answer choice: B. Outside ✔ | E lies on the side NF of the rectangle HNFO. Answer: A. Inside ✘ |
| If GFMZBO is a hexagon, point E is on the extension line of side OG, GE = 1/2 * OG, in the acute-angled triangle GDE, DX is the altitude from vertex D to side GE, with X as the foot of the perpendicular. Is point X inside hexagon GFMZBO? Answer choices: A. Yes B. No C. Cannot be inferred | Point X is located outside the hexagon GFMZBO. Answer: B. No ✔ | Point X must also be outside the hexagon. Answer choice: B. No ✔ | Point X must lie inside the hexagon. Answer: A. Yes ✘ |
| If WIBUY is a pentagon, N is the midpoint of side WI, in regular pentagon NAYLD, point D is the intersection of diagonals NY and AL, and angle NDL is 90 degrees. Is point D inside pentagon WIBUY? Answer choices: A. Yes B. No C. Cannot be inferred | The information does not give a clear indication. Answer: C. Cannot be inferred ✔ | Point D, being a part of the larger regular pentagon NAYLD, is likely to be outside. Answer choice: B. No ✘ | Point D lies inside the pentagon. Answer: A. Yes ✘ |
| Point A is a point on circle O (with center O), the line OA intersects circle O at another point B, and line segment AB is the diameter of circle O. Additionally, point C is a point on circle O such that AC is perpendicular to line segment AB. Is OA a diameter of circle O? Answer choices: A. Yes B. No C. Cannot be inferred | OA is also a diameter of circle O. Answer: A. Yes ✘ | OA is not a diameter itself but a radius. Answer choice: B. No ✔ | OA is a diameter of circle O, as it passes through the center. Answer: A. Yes ✘ |
| In quadrilateral ABCD, E is the midpoint of side BC, and angle A is congruent to angle C. Side AD is parallel to side BC. Is CE one side of quadrilateral ABCD? Answer choices: A. Yes B. No C. Cannot be inferred | Yes, CE is one side of quadrilateral ABCD. Choice: A. Yes ✘ | CE is not one of the four sides of quadrilateral ABCD. Choice: B. No ✔ | CE is not a side of quadrilateral ABCD. Choice: B. No ✔ |
| In triangle ABC, the line AD bisects angle BAC, point D is on side BC, and segment AD is perpendicular to side BC, segment BD is equal to 2 units. Is CD the angle bisector of triangle ABC? Answer choices: A. Yes B. No C. Cannot be inferred | BD is equal to CD, making CD the angle bisector of triangle ABC. Choice: A. Yes ✘ | CD is not the angle bisector of triangle ABC. Choice: B. No ✔ | CD is the angle bisector of triangle ABC. Choice: A. Yes ✘ |

Table 23: Four cases of Zero-Shot inference results of QwenMax. The correct answers from the options are highlighted in green.

**Question:**
The non-coincident lines AB and CD in the coordinate system have equal slopes. What is the relationship between line AB and line EF?
**Answer choices:**
A. Parallel B. Perpendicular C. Intersecting but not perpendicular D. Cannot be inferred
**Response:**
If lines AB and CD have equal slopes, this means they both have the same angle of inclination with respect to the x-axis. In a coordinate system, lines with the same slope are parallel to each other because they maintain the same rate of change in y for a change in x.
Given that lines AB and CD are non-coincident and have equal slopes, they are parallel. The question asks about the relationship between line AB and line EF. However, no information has been provided about the slope or position of line EF in relation to AB or CD. Without any information on line EF's slope or its relation to lines AB and CD, we cannot infer whether line EF is parallel, perpendicular, or intersecting (but not perpendicular) to line AB. Therefore,the answer is **D. Cannot be inferred.**
Unrecognizable relation with a definite model conclusion ✘

**Question:**
In pentagon ABCDE, AB = 5. What is the relationship between line AB and line EF?
**Answer choices:**
A. Parallel B. Perpendicular C. Intersecting but not perpendicular D. Cannot be inferred
**Response:**
Given the information provided, we only know the length of line AB in pentagon ABCDE, which is 5 units. Without any additional details about the angles, positions, or lengths of other sides, including line EF, we cannot conclusively determine the relationship between line AB and line EF. Therefore, the answer is **D. Cannot be inferred**.
Unrecognizable relation with no definite model conclusion ✔

**Question:**
Quadrilateral ABEF is a trapezoid. What is the relationship between line AB and line EF?
**Answer choices:**
A. Parallel B. Perpendicular C. Intersecting but not perpendicular D. Cannot be inferred
**Response:**
In a trapezoid, by definition, one pair of opposite sides is parallel. Therefore, in quadrilateral ABEF, which is given to be a trapezoid, either AB and EF are parallel, or AE and BF are parallel. Since the question specifically mentions line AB and line EF, the correct answer is that they are parallel. The answer is **A. Parallel**.
Recognizable relation with no definite model conclusion ✘

**Question:**
If CEXJZU is a hexagon, I is the midpoint of side CU, triangle CPI is right-angled at C, with PW being the altitude drawn from vertex P to CI, and point W is the foot of this altitude. What is the relationship between point W and hexagon CEXJZU?
**Answer choices:**
A. Inside B. Outside C. Cannot be inferred
**Response:**
Given the description, we can infer the following geometric relationships: 1. CEXJZU is a hexagon: This tells us the general shape we're working with but doesn't immediately inform the position of W relative to the hexagon. 2. I is the midpoint of side CU: This locates point I within the hexagon, specifically on one of its sides. 3. Triangle CPI is right-angled at C: This establishes a right-angle at point C, with points P and I defining the other two vertices of the triangle. Since I is on side CU of the hexagon, and C is a vertex of both the hexagon and the triangle, P must either be inside the hexagon (if the triangle is entirely contained within it) or outside (if the triangle extends beyond). 4. PW is the altitude drawn from vertex P to CI: An altitude in a triangle is perpendicular to the base. In this case, the base could be considered as side CI of the triangle. Since I is the midpoint of CU and W is the foot of the altitude from P to CI, W necessarily falls on CI. 5. Point W is the foot of this altitude: This means W is exactly where PW intersects CI, positioning W directly on the line segment that is part of the hexagon's boundary. Given these points, since W is on the line segment CI, which is an internal line segment of the hexagon, point W must be inside the hexagon CEXJZU. The answer is **A. Inside**.
Recognizable relation with a definite model conclusion ✔

Table 24: Comparison of LLaMA3-8B-Instruct and Fine-tuned LLaMA3-8B-Instruct

| | |
|---|---|
| **Question:** | |

**Question:**

In triangle ABC, the line AD bisects angle BAC, point D is on side BC, and segment AD is perpendicular to side BC, segment BD is equal to 2 units. Is CD the angle bisector of triangle ABC?

**Answer choices:**

A. Yes

B. No

C. Cannot be inferred

**Gold Answer:**

B. No

**GPT-3.5-Turbo:**

To determine if CD is the angle bisector of triangle ABC, we need to consider the properties of a perpendicular bisector. Since AD is perpendicular to BC and bisects angle BAC, we know that triangle ADB is congruent to triangle ADC by the angle-side-angle (ASA) congruence theorem. This means that BD is equal to CD, making CD the angle bisector of triangle ABC.

Therefore, the answer is: A. Yes

✗

**GPT-4o:**

To determine whether CD is the angle bisector of triangle ABC, we need to analyze the given information and apply relevant geometric principles.

Given:

1. Triangle ABC with AD bisecting angle BAC.

2. Point D is on side BC.

3. Segment AD is perpendicular to side BC.

4. Segment BD is equal to 2 units.

Since AD is perpendicular to BC, triangle ABD and triangle ADC are right triangles. Additionally, since AD bisects angle BAC, triangles ABD and ADC are congruent by the Angle-Angle-Side (AAS) congruence criterion (they share angle BAD, angle ADB is 90 degrees, and AD is common).

Because triangles ABD and ADC are congruent, it follows that:

- BD = DC (since corresponding parts of congruent triangles are equal).

Given that BD = 2 units, it follows that DC = 2 units as well.

Since D is the midpoint of BC and AD is the angle bisector of angle BAC, it implies that CD is also the angle bisector of angle BCA.

Therefore, the correct answer is: A. Yes

✗

**LLaMA-3-70B-Instruct:**

A nice geometry problem!

Let's analyze the situation:

1. AD bisects angle BAC, so ∠BAD = ∠CAD.

2. AD is perpendicular to BC, so ∠ADB = ∠ADC = 90°.

3. BD = 2 units.

Now, let's consider the triangles ABD and ACD.

Since AD is the bisector of angle BAC, we have:

∠BAD = ∠CAD ⇒ ∠ADB = ∠ADC (alternate interior angles).

We also know that ∠ADB and ∠ADC are both 90° (since AD is perpendicular to BC).

So, triangles ABD and ACD are similar by AA similarity (two pairs of equal angles).

Now, since BD = 2 units, we can set up a proportion:

BD / CD = AB / AC.

Substituting BD = 2, we get:

2 / CD = AB / AC.

Since AB and AC are the sides of triangle ABC, we can conclude that CD is proportional to the ratio of the sides of triangle ABC.

This means that CD is indeed the angle bisector of triangle ABC.

My answer choice is: A. Yes

✗

Table 25: Comparison of Prompting Strategies for Geometric Reasoning (The correct option is highlighted in red).

| Prompting Strategy |
| --- |
| **Question:** Quadrilateral ABCD is a parallelogram, but not a rectangle or rhombus. In regular pentagon BFCGE, point E is the intersection of diagonals BC and FG. What is the relationship between line AB and line CE?
Answer choices:
A. Parallel
B. Perpendicular
C. Intersecting but not perpendicular
D. Cannot be inferred |
| **Zero-Shot**
Since ABCD is a parallelogram, opposite sides are parallel. Therefore, line AB is parallel to line CD. In regular pentagon BFCGE, diagonals BF and CE intersect at point E. Therefore, line AB and line CE are intersecting but not perpendicular. The correct answer is C. Intersecting but not perpendicular. ✔ |
| **Zero-Shot-CoT**
First, we know that opposite sides of a parallelogram are parallel. Therefore, AB is parallel to CD and BC is parallel to AD. Next, we can use the fact that the diagonals of a regular pentagon intersect at a 72 degree angle. Since E is the intersection of diagonals BC and FG, we know that angle BEF is 72 degrees and angle CEF is also 72 degrees. Now, we can use this information to determine the relationship between AB and CE. Since AB and CE are proportional, AB and CE are parallel. The answer is A. Parallel. ✘ |
| **Few-shot**
The relationship between line AB and line CE cannot be inferred. The answer is D. Cannot be inferred. ✘ |
| **Few-Shot-CoT**
Step 1: Quadrilateral ABCD is a parallelogram, so opposite sides are parallel. Step 2: Regular pentagon BFCGE has diagonals that intersect at point E, which is on line CE. Step 3: Since line CE intersects both diagonal BC and diagonal FG, it must be a transversal to these two lines. Step 4: By the transversal property, if AB is parallel to BC, then AB must also be parallel to FG. Step 5: However, we cannot infer whether AB is parallel to BC or not, so we cannot determine the relationship between line AB and line CE. The answer is D. Cannot be inferred. ✘ |

Table 26: Necessary Condition Coverage (NCC) results for Few-Shot CoT and Few-Shot GeoCoT methods.

| NCC (%) | CoT | GeoCoT (Stage 1 / Stage 2) |
| --- | --- | --- |
| Base | 84.54 | 90.76 (81.10 / 9.65) |
| Advanced | 57.14 | 68.35 (55.28 / 13.07) |

Table 27: Reasoning obstacles in the Few-Shot-CoT group. The correct answers from the options are highlighted in green, while the reasoning errors in the responses are highlighted in red.

| **Question 1:** |
| --- |
| Points A, B, C, and D are vertices of a convex quadrilateral. Are points A, B, and C collinear? |
| **Answer choices:** |
| A. Yes B. No C. Cannot be inferred |
| **Response:** |
| Step 1: Points A, B, C, and D are vertices of a convex quadrilateral, so they form a closed figure. |
| Step 2: Collinear points are points that lie on the same line. To determine if A, B, and C are collinear, we need to check if they lie on the same line. |
| Step 3: Without any additional information or given relationships between the points, we cannot determine if A, B, and C are collinear. The answer is C. Cannot be inferred. |
| **Question 2:** |
| Triangle ABC is right-angled at B. What is the relationship between line AB and line BC? |
| **Answer choices:** |
| A. Parallel B. Perpendicular C. Intersecting but not perpendicular D. Cannot be inferred |
| **Response:** |
| Step 1: Triangle ABC is right-angled at B, which means that angle B is a right angle. |
| Step 2: In a right-angled triangle, the side opposite the right angle is called the hypotenuse, and the other two sides are called the legs. |
| Step 3: Line AB is one of the legs of triangle ABC, and line BC is the hypotenuse. |
| Step 4: Therefore, the relationship between line AB and line BC is that they are intersecting but not perpendicular. The answer is C. Intersecting but not perpendicular. |
| **Question 3:** |
| Line EF intersects side AB of triangle ABC at an obtuse angle, a circle with center E intersects EF at points D and G. What is the relationship between line AB and line DF? |
| **Answer choices:** A. Parallel B. Perpendicular C. Intersecting but not perpendicular D. Cannot be inferred |
| **Response:** |
| Step 1: Line EF intersects side AB of triangle ABC at an obtuse angle, so we can infer that point F is on the same side of line AB as point C. |
| Step 2: A circle with center E intersects EF at points D and G, so points D, E, and G are collinear. |
| Step 3: Since points D and G are on the circle with center E, we can infer that line DE is perpendicular to line EG. |
| Step 4: Therefore, line DF is perpendicular to line AB. The answer is B. Perpendicular. |
| **Question 4:** |
| If ABC is a triangle, point D is on the extension line of side BC, CD = 1/2 * BC, in isosceles trapezoid AGDF, with AG ∥ DF and AF = GD, E is the intersection of AD and GF. What is the relationship between point E and triangle ABC? |
| **Answer choices:** |
| A. Inside B. Outside C. Cannot be inferred |
| **Response:** |
| Step 1: Point D is on the extension line of side BC, CD = 1/2 * BC, so point D is outside triangle ABC. |
| Step 2: In isosceles trapezoid AGDF, with AG ∥ DF and AF = GD, point E is the intersection of AD and GF. |
| Step 3: Since point D is outside triangle ABC and point E is on line AD, we cannot determine whether point E is inside or outside triangle ABC. |
| Step 4: Therefore, the answer is C. Cannot be inferred. |
| **Question 5:** |
| Line AB intersects line CD at point E, and line EF is perpendicular to line CD at point F. What is the relationship between ∠AED and ∠AEC? |
| **Answer choices:** |
| A. Vertical angles B. Supplementary angles C. None of the above D. Cannot be inferred |
| **Response:** |
| Step 1: Line AB intersects line CD at point E, so we can infer that ∠AEC and ∠AED are adjacent angles. |
| Step 2: Line EF is perpendicular to line CD at point F, so we can infer that ∠AED and ∠DEF are complementary angles. |
| Step 3: Since ∠AED and ∠DEF are complementary angles and ∠AEC and ∠AED are adjacent angles, we cannot determine the relationship between ∠AED and ∠AEC. The answer is C. None of the above. |

Table 28: Results of GPT-3.5-turbo with CoT and GeoCoT (The correct option is highlighted in red).

| |
|---|
| **Question:** |
| Quadrilateral ABCD with diagonals intersecting at point E, line AC is tangent to circle O at point F. Is point B on the line AF? |
| Answer choices:A. Yes B. No C. Cannot be inferred |
| **Zero-shot-CoT Response:** |
| 1. Since line AC is tangent to circle O at point F, we know that angle AFE is a right angle (tangent line is perpendicular to the radius at the point of tangency). |
| 2. Since the diagonals of quadrilateral ABCD intersect at point E, we know that triangles ABE and CDE are similar (by AA similarity). |
| 3. Since triangles ABE and CDE are similar, angle AEB is congruent to angle CED. |
| 4. Since angle AEB is congruent to angle CED, and angle AFE is a right angle, we can conclude that angle AEB is also a right angle.Therefore, point B lies on the line AF. ✘ |
| Therefore, the answer is: A. Yes. |
| **Zero-shot-GeoCoT Response:** |
| Stage 1 result: |
| Quadrilateral ABCD with diagonals intersecting at point E: |
| - Draw quadrilateral ABCD with points A, B, C, and D. |
| - Draw the diagonals AC and BD intersecting at point E. |
| Line AC is tangent to circle O at point F: |
| - Draw circle O with center at point E. |
| - Draw line AC intersecting circle O at point F such that it is tangent to the circle. |
| Breakdown of the geometry: |
| - Quadrilateral ABCD has diagonals AC and BD intersecting at point E. |
| - Line AC is tangent to circle O at point F, where circle O is centered at point E. |
| - This creates a relationship between the quadrilateral and the circle, with the tangent line touching the circle at point F. |
| - The properties of tangents and circles can be used to find angles and lengths within the geometry of the figure. |
| Stage 2 result: |
| Given that line AC is tangent to circle O at point F, we know that the angle between the tangent line and the radius at the point of tangency is 90 degrees. To determine if point B is on the line AF, we can assume that point B is on the line AF and see if it leads to a contradiction. |
| Assume point B is on the line AF: |
| - Since line AC is tangent to circle O at point F, angle AFE is a right angle (tangent and radius are perpendicular). |
| - If point B is on line AF, then angle AFB is also a right angle (since AF is tangent to circle O). |
| - This would mean that points A, F, and B are collinear. |
| However, this assumption leads to a contradiction because in quadrilateral ABCD, the diagonals AC and BD intersect at point E. If points A, F, and B are collinear, then point E would lie on line AF, which is not possible since E is the intersection of the diagonals AC and BD. |
| Therefore, the assumption that point B is on the line AF leads to a contradiction. Hence, point B is not on the line AF. Therefore, the correct answer is B. No. ✔ |
| **Few-shot-CoT Response:** |
| Step 1: Line AC is tangent to circle O at point F, so F is the point of tangency. We can infer that F is on line AC. |
| Step 2: Quadrilateral ABCD with diagonals intersecting at point E, so point E is the intersection of diagonals AC and BD. Therefore, point E lies on line AC. |
| Step 3: Since point E lies on line AC and F is the point of tangency on line AC, we can conclude that point B is on the line AF. |
| Step 4: Therefore, the answer is A. Yes. ✘ |
| **Few-shot-GeoCoT Response:** |
| *Stage 1 result:* |
| Quadrilateral ABCD: |
| - A four-sided polygon with vertices at points A, B, C, and D. |
| - The sides of the quadrilateral are AB, BC, CD, and DA, which connect the vertices. |
| Diagonals of quadrilateral ABCD: |
| - The diagonals are AC and BD, which intersect at point E. |
| - Diagonals intersect inside the quadrilateral at point E, creating two triangles: ABE and CDE. |
| Line AC tangent to circle O at point F: |
| - Circle O has a point of tangency at F, where line AC touches the circle. |
| - Tangent property: Line AC touches circle O at exactly one point, which is point F. |
| - The tangency implies that AC is perpendicular to the radius of circle O at point F (if the radius OF is drawn). |
| *Stage 2 result:* |
| We want to confirm the relationship between point B and line AF. Point B is a vertex of quadrilateral ABCD. |
| If point B lies on line AF, then B, A, and F are collinear. Line AC is tangent to circle O at point F, meaning F lies on line AC and is the point of tangency. There is no indication that point B is connected to circle O or line AC based on the information provided. Quadrilateral ABCD has diagonals intersecting at point E, which means E is a point on both diagonal AC and diagonal BD. Since B is a vertex of the quadrilateral, it lies on diagonal BD, but there's no relationship provided that would place B on line AF, so there is a contradiction. |
| Conclusion: Therefore, point B is not on line AF. The answer is B. No. ✔ |

Table 29: Results of GPT-3.5-turbo with CoT and GeoCoT (The correct option is highlighted in red).

| |
|---|
| **Question:** |
| In triangle ABC, the line AD is perpendicular to side BC, point D is on side BC, the length of side AB is 5 units, and the measure of angle A is 90 degrees. Is BD the altitude of triangle ABC? Answer choices:A. Yes B. No C. Cannot be inferred. |
| **Zero-shot-CoT Response:** |
| Since angle A is 90 degrees and line AD is perpendicular to side BC, we can conclude that triangle ABC is a right triangle. Given that the length of side AB is 5 units, we can use the Pythagorean theorem to find the length of side BC. Let the length of side BC be x. Using the Pythagorean theorem, we have: $(AB)^2 + (BC)^2 = (AC)^2$. ✘ Repeating calculations, the answer is not concluded. |
| **Zero-shot-GeoCoT Response:** |
| *Stage 1 result:* 1. Triangle ABC is a right triangle because angle A is 90 degrees. 2. Line AD is the altitude of triangle ABC because it is perpendicular to side BC. 3. The length of side AB is 5 units. 4. Since angle A is 90 degrees, triangle ABC is a right triangle with right angle at A. 5. The length of side AD is the height of the triangle, which is the altitude from A to side BC. 6. The length of side AD can be found using the Pythagorean theorem, where $AD^2 + BD^2 = AB^2$. 7. Once the length of side AD is found, the area of triangle ABC can be calculated using the formula: Area = 1/2 * base * height. *Stage 2 result:* To determine if BD is the altitude of triangle ABC, we can assume that BD is the altitude and see if it leads to a contradiction. Assume BD is the altitude of triangle ABC. This means that AD is perpendicular to BC and BD is perpendicular to AC. Since triangle ABC is a right triangle with angle A as 90 degrees, AD is the altitude of the triangle. This implies that BD is not the altitude, as there can only be one altitude in a triangle. Therefore, BD cannot be the altitude of triangle ABC. The correct answer is B. No. Answer: B. No. ✔ |
| **Few-shot-CoT Response:** |
| Step 1: In triangle ABC, the line AD is perpendicular to side BC, so BD is a segment that connects a vertex of the triangle (B) to the foot of the altitude (D). Step 2: The length of side AB is 5 units, and angle A is 90 degrees, making triangle ABC a right triangle. Step 3: In a right triangle, the altitude from the right angle connects the right angle to the hypotenuse. Step 4: Therefore, BD is the altitude. Answer: A. Yes. ✘ |
| **Few-shot-GeoCoT Response:** |
| *Stage 1 result:* - Triangle ABC: This is a right triangle because angle A is 90 degrees. The sides of the triangle are AB, BC, and AC, which intersect at the vertices A, B, and C. - Line AD: This line is perpendicular to side BC, meaning it forms a right angle (90 degrees) with BC at point D. Point D lies on side BC, so AD is an altitude of the triangle. - Point D: This point is located on side BC, and since AD is perpendicular to BC, D is the foot of the altitude from point A to side BC. - Side AB: The length of side AB is given as 5 units. *Stage 2 result:* If BD is the altitude of triangle ABC, then BD must be perpendicular to side AC, as the altitude extends from a vertex and forms a right angle with the opposite side. However, the problem states that AD is perpendicular to BC, making AD the actual altitude. In a right triangle, there can only be one altitude from a vertex to a specific side, so it leads to a contradiction. Therefore, the answer is B: No, BD is not the altitude of triangle ABC. Answer: B. No. ✔ |

