# OpenReview forum: "Do Large Language Models Truly Understand Geometric Structures?"
_ICLR.cc/2025/Conference — ICLR 2025 Poster_

### Official Review · Reviewer_8Tsz · 2024-10-30

**Soundness:** 3
**Presentation:** 3
**Contribution:** 3
**Rating:** 6
**Confidence:** 3

**Summary:**

This paper investigates LLMs abilities to understand geometric relationships and spatial reasoning based on textual descriptions. The authors observe that LLMs often arrive at correct answers without grasping the underlying geometric relationships. To address this, they propose a new task called Geometric Relationship Identification (GRI) and introduce GeomRel, a dataset specifically designed to evaluate models' abilities in GRI. For dataset construction, they create a collection of basic geometric relationships and develop more complex examples by combining and enhancing these foundational relationships.

The authors conduct extensive experiments across several models and strategies, including zero-shot, few-shot, Chain-of-Thought (CoT), and fine-tuning methods. Their findings indicate that current LLMs have significant limitations in spatial understanding and that most strategies yield limited improvements. They propose a new prompting technique, GeoCoT, which decomposes geometric observations and employs reverse reasoning.

**Strengths:**

* The paper establishes GRI as a new task, shifting focus beyond answer accuracy to intermediate steps in spatial reasoning.
* The proposed dataset, GeomRel, is valuable, especially with its advanced split that incorporates logical chains, indeterminate cases, and extraneous information, mimicking real-world problem complexity and ambiguity.
* The paper provides a thorough evaluation across multiple LLMs and reasoning methods, yielding insights into their spatial reasoning capacities and limitations.

**Weaknesses:**

* For disambiguation, the authors manually reviewed and excluded ambiguous data throughout the data construction process, which may reduce scalability and limit others’ ability to expand the benchmark.
* The paper’s focus on GRI as an isolated skill may be too narrow, leaving it uncertain if success in GRI tasks will translate to general spatial reasoning or even multi-step problem-solving abilities.
* The two-stage GeoCoT method, which involves decomposing geometry problems and reverse reasoning, may be challenging to scale to more complex tasks.

**Questions:**

* While the advanced split of GeomRel does require managing multiple geometric relationships, it still treats each relationship as relatively independent. Real-world geometric problems, however, often demand that LLMs execute step-by-step calculations in sequence, where each conclusion logically affects the next.
* Some analyses in Appendix F are particularly insightful, demonstrating the reasoning path and showing how the vanilla model fails while reverse reasoning aids understanding. It would be valuable to provide experimental results to support these explanations.
*  the paper show marginal gains from fine-tuning. It would be valuable to explore the reasons behind this limited improvement, possibly factors like dataset size, or the nature of the fine-tuning data.
* In explaining the performance improvements from point relabeling, the authors hypothesize that "using more complex descriptions may stimulate LLMs’ reasoning abilities." This point could benefit from further explanation.

---

> ### Author Response · Authors · 2024-11-19
> **Response 1: Additional Results and Explanations**
>
> Thanks for your constructive feedback. We respond to your questions one by one:
>
> >Q1: While the advanced split of GeomRel does require managing multiple geometric relationships, it still treats each relationship as relatively independent. Real-world geometric problems, however, often demand that LLMs execute step-by-step calculations in sequence, where each conclusion logically affects the next.
>
> Thank you for your thorough reading and deep reflection on our paper. Our evaluation separates relationships into distinct categories: **line-based relations** focus on fine-grained interactions between points and lines, **angle-based relations** involve directional and symmetry-related properties, and **shape-based relations** assess holistic understanding of geometric structures like circles and polygons.
>
> While complex problems often involve interactions between relationships, this decomposition helps analyze specific skills like local understanding and overall positional reasoning. We also limit reasoning chains to 2–3 steps for advanced tasks to better evaluate core reasoning capabilities.
>
> >Q2: Some analyses in Appendix F are particularly insightful, demonstrating the reasoning path and showing how the vanilla model fails while reverse reasoning aids understanding. It would be valuable to provide experimental results to support these explanations.
>
> We appreciate your positive feedback on the analyses in Appendix F. To further support our explanations, we introduced a new metric, **Necessary Condition Coverage (NCC)**, which measures the proportion of necessary intermediate conditions covered during the reasoning process that lead to the correct answer. We evaluated this on 24 random selected questions using GPT-3.5-turbo with Few-Shot CoT and Few-Shot GeoCoT methods. The results are as follows:
>
> | NCC (%)       | CoT   | GeoCoT (Stage 1 / Stage 2)  |
> |---------------|-------|-|
> | Base          | 84.54 | 90.76 (81.10 / 9.65)       |
> | Advanced      |57.14 |68.35 (55.28 / 13.07) |
>
> The results demonstrate that our two-stage GeoCoT method improves NCC, particularly for advanced problems. This highlights the effectiveness of decomposing geometry problems and incorporating reverse reasoning to enhance coverage and reasoning quality.
>
>
>
> >Q3: the paper show marginal gains from fine-tuning. It would be valuable to explore the reasons behind this limited improvement, possibly factors like dataset size, or the nature of the fine-tuning data.
>
> Thank you for your suggestion. We conducted further analysis on the fine-tuning results of LLaMA3-8B-Instruct and found that its marginal gains were due to the model frequently outputting the **uncertain answer**, "cannot be inferred." Additionally, we fine-tuned the LLaMA3-8B-Base model, which showed substantial improvement, achieving **81.26%** and **62.75%** accuracy on the Basic and Advanced subsets, respectively, surpassing GPT-4o.
>
> We analyzed the proportion of "cannot be inferred" responses generated by the models, as shown below (with *Truth* representing the actual proportion in the dataset):
>
> |Model|Basic|Advanced|
> |-|--------|----------|
> |LLaMA3-8B-Instruct|14.71%|8.21%|
> |Fine-tunedLLaMA3-8B-Instruct|71.53%|67.40%|
> |Fine-tunedLLaMA3-8B-Base|25.03%|21.41%|
> |Truth|27.94%|26.49%|
>
> The fine-tuned LLaMA3-8B-Instruct model heavily leaned toward uncertain answers, whereas the Fine-tuned LLaMA3-8B-Base model closely aligned with the actual data distribution. This discrepancy highlights the impact of fine-tuning data and the model's initial instruction-tuning bias.
>
> >Q4: In explaining the performance improvements from point relabeling, the authors hypothesize that "using more complex descriptions may stimulate LLMs’ reasoning abilities." This point could benefit from further explanation.
>
>
> Thank you for raising this point. Initially, point representations used simpler labels such as "triangle ABC." Through **Re-labeling Points (RP),** we introduced labels like "triangle MGQ," which, while preserving the geometric structure, add complexity to the description. We evaluated this approach on the **base subset** of the dataset, and the results are as follows:
>
> |Subset|Original|AfterRP|
> |------------|--------------|--------------|
> |Line|63.16|73.38(+10.26)|
> |Angle|46.37|49.62(+3.25)|
> |Shape|65.79|68.42(+2.63)|
>
> The improvement suggests that Re-labeling Points stimulates the model to adopt more complex reasoning paths for problems that initially appeared simpler. However, this effect diminishes for the **advanced subset,** where descriptions are already sufficiently complex. This indicates that while RP enhances performance by unlocking latent reasoning potential, it does not fundamentally improve the model's reasoning ability.

---

> > ### Author Response · Authors · 2024-11-19
> > **Response 1: Explanations for the Limitations**
> >
> > >W1 & W2:  The manual disambiguation process may limit scalability and the benchmark’s expansion, and focusing on GRI as an isolated skill raises questions about its applicability to broader spatial reasoning or multi-step problem-solving tasks.
> >
> > We acknowledge that manual disambiguation is indeed a labor-intensive process, but it is necessary due to the nature of geometric problems. Ensuring the dataset contains **clear and valid geometric structures** requires rigorous checking and refinement. To address concerns about scalability, we plan to release a detailed **disambiguation guide** outlining our review process. This will facilitate future efforts to expand or adapt the benchmark.
> >
> >
> > >W3: The two-stage GeoCoT method, which involves decomposing geometry problems and reverse reasoning, may be challenging to scale to more complex tasks.
> >
> > You are correct that our two-stage GeoCoT method is tailored specifically for geometric problems and may not be directly transferable to other tasks. However, the **core idea of combining decomposition with reverse reasoning** is conceptually adaptable. For instance, in domains like physics or circuit analysis, problems could be broken down into subproblems involving individual components, while reverse reasoning could validate intermediate results.
> >
> > While scaling to more complex tasks presents challenges, we believe that refining and generalizing these principles could provide valuable insights for broader applications in structured reasoning tasks.

---

> > > ### Author Response · Authors · 2024-11-24
> > > **Look forward to feedbacks**
> > >
> > > Dear Reviewer 8Tsz,
> > >
> > > We first thank you again for your constructive comments. We have addressed your concerns one by one and supplemented more detailed experimental results and explanations. We look forward to further discussion with you and your positive feedback about our rebuttal.
> > >
> > > Best regards,
> > >
> > > Authors

---

### Official Review · Reviewer_Nc3K · 2024-11-04

**Soundness:** 3
**Presentation:** 3
**Contribution:** 3
**Rating:** 8
**Confidence:** 3

**Summary:**

This paper addresses the limitations of large language models (LLMs) in understanding geometric structures, proposing a new benchmark, GeomRel, specifically designed to assess LLMs' ability to identify geometric relationships. The authors identify that while existing datasets mainly measure final answer accuracy, they fail to capture whether LLMs truly understand underlying geometric structures. The GeomRel benchmark isolates the task of geometric relationship identification (GRI) as a foundational skill for geometric reasoning. Using GeomRel, the paper evaluates several LLMs and finds that most perform well on simple geometric relationships but struggle with more complex structures, particularly those involving angle relationships. The authors introduce a new method, Geometry Chain-of-Thought (GeoCoT), which decomposes geometric reasoning into step-by-step relationship identification, significantly improving model performance on GeomRel by over 9% on basic tasks and nearly 15% on advanced tasks.

**Strengths:**

1. The GeomRel dataset, which isolates geometric relationship identification as a key step, provides a novel and focused way to evaluate LLMs' geometric understanding. This dataset addresses a unique gap in the field and enables more focused evaluation of geometric reasoning capabilities.

2. The Geometry Chain-of-Thought (GeoCoT) method improves LLMs’ performance in identifying geometric relationships by breaking down problems into reasoning steps. The proposed method significantly improved both basic and advanced variants of the dataset.

3. The authors evaluates a range of LLMs, including both proprietary and open-source models, providing a rounded view of how different models handle geometric reasoning. This extensive evaluation is valuable for identifying model-specific strengths and weaknesses in geometric tasks.

4. The paper investigates multiple prompting methods, including Zero-Shot and Chain-of-Thought (CoT), analyzing their effectiveness in GRI tasks. This analysis highlights that while traditional CoT is less effective for geometric tasks, the GeoCoT adaptation provides usefgul improvements for geometric reasoning.

**Weaknesses:**

1. Although fine-tuning on GeomRel was attempted, the results only report results of a single model. More experiments exploring different models could clarify which models has performance improvements with fine-tuning.
2. It is difficult to truly understand the difficulty of the dataset for human intelligence. This can be understood by sampling the dataset and carrying out a systematic human evaluation to report human baselines on these geometrics reasoning problems.
3. GeomRel, while valuable for evaluating LLMs on basic geometric relationships, covers a limited range of geometric concepts, primarily focusing on 2D relationships involving lines, angles, and shapes. This narrow scope may not fully represent the complexity of real-world geometric tasks and it will further help evaluate reasoning capabilities particularly those that involve 3D relationships, transformations (like rotations and reflections), or coordinate-based reasoning.

**Questions:**

1. It would help the readers better, if the authors can provide the exact versions of proprietary LLMs that was used as well as other hyper parameters set for these models for inferences as well as for the fine tuning experiments.

---

> ### Author Response · Authors · 2024-11-19
>
> >W1: Although fine-tuning on GeomRel was attempted, the results only report results of a single model. More experiments exploring different models could clarify which models has performance improvements with fine-tuning.
>
> First, thanks very much for your recognition of our work and the suggestions. We have added the fine-tuning results for the LLaMA3-8B-Base model, as shown below:
>
> |Model|Line-based (basic)|Line-based (advanced)|Angle-based (basic)|Angle-based (advanced)|Shape-based (basic)|Shape-based (advanced)|
> |-----------------------------|--------------------|-----------------------|---------------------|------------------------|---------------------|------------------------|
> |LLaMA3-8B-Instruct|63.16|42.11|52.17|13.04|26.92|38.46|
> |FinetunedLLaMA3-8B-Instruct|34.14|40.56|32.50|13.75|39.87|40.51|
> |FinetunedLLaMA3-8B-Base|**86.86**|**71.67**|63.16|**54.90**|**93.75**|**61.69**|
> |GPT-4o|77.87|52.91|**66.67**|29.00|87.04|53.38|
>
> The results show that fine-tuning on the LLaMA3-8B-Base model is highly effective, significantly improving accuracy across all subsets. It also outperforms the best-performing model (GPT-4o) in most categories. Comparing the two fine-tuned models, we found that the Finetuned LLaMA3-8B-Instruct model underperforms, likely due to its tendency to select the uncertain option "cannot be inferred." This additional experiment highlights the importance of model selection for fine-tuning on GeomRel. Additionally, fine-tuning experiments on other models (e.g., the Qwen series) are ongoing, and we will provide updated results later.
>
>
>
> >W2: It is difficult to truly understand the difficulty of the dataset for human intelligence. This can be understood by sampling the dataset and carrying out a systematic human evaluation to report human baselines on these geometrics reasoning problems.
>
> Thank you for this valuable suggestion. To address this, we conducted an evaluation with five graduate students in computer science. The results, along with comparisons to GPT-4o and random baselines, are shown below:
>
> | Model       | Line-based  (Basic) | Line-based (Advanced) | Angle-based (Basic) | Angle-based (Advanced) | Shape-based (Basic) | Shape-based (Advanced) | Average (Basic) | Average (Advanced) |
> |-------------|------------------|---------------------|-------------------|----------------------|-------------------|----------------------|---------------|------------------|
> | GPT-4o      | **77.87**           | **52.91**              | **66.67**            | 29.00               | 87.04            | 58.38               | **77.86**         | **47.93**           |
> | Random      | 28.53           | 29.43              | 25.00            | 20.00               | 33.33            | 33.33               | 28.95         | 27.59           |
> | Human       | 71.73           | 39.34              | 52.86            | **34.63**              | **90.63**            | **69.41**              | 71.74         | 47.79           |
>
> We found that human performance is overall similar to GPT-4o, but with notable differences across subsets. GPT-4o outperforms humans on **Line-based** and **Angle-based** subsets, indicating stronger capabilities in fine-grained reasoning. However, humans significantly outperform GPT-4o on **Shape-based** tasks, suggesting that the model may struggle with holistic perception of geometric structures compared to humans.
>
>
> >W3: GeomRel, while valuable for evaluating LLMs on basic geometric relationships, covers a limited range of geometric concepts, primarily focusing on 2D relationships involving lines, angles, and shapes. This narrow scope may not fully represent the complexity of real-world geometric tasks and it will further help evaluate reasoning capabilities particularly those that involve 3D relationships, transformations (like rotations and reflections), or coordinate-based reasoning.
>
> Thank you for highlighting this limitation. Our work intentionally focuses on 2D geometric relationships to ensure thorough coverage within this scope, as well as to align with the current reasoning capabilities of LLMs. While 3D relationships and transformations were considered during dataset design, we chose to prioritize foundational 2D concepts. Expanding GeomRel to include 3D reasoning and transformations is an important direction for future work as model capabilities continue to improve.
>
> >Q1: It would help the readers better, if the authors can provide the exact versions of proprietary LLMs that was used as well as other hyper parameters set for these models for inferences as well as for the fine tuning experiments.
>
> Thank you for your suggestion. We have added the exact versions of the proprietary LLMs and the details of the hyperparameters used for inference and fine-tuning experiments to the appendix of the paper.

---

> > ### Author Response · Authors · 2024-11-24
> > **Supplementary Fine-Tuning Experiments Results**
> >
> > Adding to the previous results, we conducted fine-tuning experiments on the Qwen2-7B models, with detailed outcomes shown below:
> >
> > | Model                       | I.Line-based (basic/advanced) | II.Angle-based (basic/advanced) | III.Shape-based (basic/advanced) |
> > |-----------------------------|-----------------------------|------------------------------|------------------------------|
> > | LLaMA3-8B-Instruct          | 63.16 / 42.11              | 52.17 / 13.04               | 26.92 / 38.46               |
> > | LLaMA3-8B-Instruct-FT| 34.14 / 40.56              | 32.50 / 13.75               | 39.87 / 40.51               |
> > | **LLaMA3-8B-Base-FT**    | 86.86 / **71.67**          | 63.16 / **54.90**           | 93.75 / 61.69           |
> > | Qwen2-7B-Instruct           | 65.55 / 31.84               | 53.75 / 20.71                 | 55.38 / 34.69                |
> > |**Qwen2-7B-Instruct-FT** | 82.14 / 70.87              | **78.95** / 54.44               | 94.48 / 68.83              |
> > | **Qwen2-7B-Base-FT**     | **89.29** / 70.48              | 73.68 / 51.51               | **95.70** / **70.13**              |
> > | GPT-4o                      | 77.87 / 52.91              | 66.67 / 29.00               | 87.04 / 53.38               |
> >
> > The results highlight that fine-tuning on both Qwen2-7B-Base and Qwen2-7B-Instruct achieves significant improvements, with the Base model slightly outperforming the Instruct variant in most cases. Compared to LLaMA3-8B-Instruct, Qwen2-7B-Instruct fine-tuning demonstrates strong results, suggesting that differences in instruction tuning methods and pretraining data may influence post-fine-tuning performance. We have included these supplementary results (along with the experimental setup) in the appendix.
> >
> > We sincerely look forward to engaging in further discussions.

---

> > > ### Comment · Reviewer_Nc3K · 2024-12-03
> > >
> > > Dear authors, thank you for addressing my questions and suggestions. I had already provided a score of 8, so I will be keeping my score.

---

### Official Review · Reviewer_XLWM · 2024-11-10

**Soundness:** 3
**Presentation:** 3
**Contribution:** 2
**Rating:** 6
**Confidence:** 4

**Summary:**

This paper investigates whether large language models can truly understand geometric structure and solve geometric problems. It introduces a new benchmark, **GeomRel**, focused on geometric relationships in three categories: line-based, angle-based, and shape-based relationships. To expand the dataset’s depth and complexity, the authors construct advanced and diverse data subsets within GeomRel. Evaluations on this benchmark reveal that even advanced models like GPT-4o outperform Random Guess by only 20.34% on complex tasks in the advanced GeomRel subset. Building on insights from these evaluations, the paper proposes a two-stage pipeline that guides models to decompose geometric structures and perform relationship observation. Experimental results demonstrate the effectiveness of the proposed pipeline in enhancing model performance on geometric tasks.

**Strengths:**

- This paper introduces a new benchmark that reveals current LLMs struggle to effectively recognize geometric structures.
- It demonstrates that even with few-shot prompting or fine-tuning, current LLMs do not perform well in recognizing geometric relationships. To address this, the paper proposes a two-stage pipeline that guides LLMs to decompose and observe geometric structures.
- Experimental results in the final section show that the proposed two-stage pipeline effectively enhances LLM performance on geometric tasks.

**Weaknesses:**

- This paper dedicates significant space to describing the rules and various data augmentation methods used in constructing the benchmark. However, the overall process and rationale for construction could be presented more clearly.
- In Section 3.5, the paper uses the LLaMA-3-8B-Instruct model as the base model for fine-tuning, which is somewhat unconventional, as it is more typical to fine-tune base models on math-related datasets.
- Including additional experimental metrics commonly used in math domains, such as pass@k, majority voting, or best-of-n, could improve the depth of analysis and provide a more comprehensive evaluation of model performance in this paper.

**Questions:**

- Could you elaborate on how the benchmark was constructed and clarify how the different relationships, difficulty levels, and diversity augmentation methods were integrated? And for instance do you leverage LLMs to help generate the benchmark or all the questions are based on rules?
- Could you provide results from fine-tuning on the LLaMA-3-8B base model?

---

> ### Author Response · Authors · 2024-11-19
>
> Thanks for your constructive feedback. We respond to your questions one by one:
>
> >W1 & Q1: Clarification on how the benchmark was constructed, including relationships, difficulty levels, and augmentation methods, and whether LLMs were involved.
>
> Thank you for your insightful question. The benchmark construction begins with fundamental geometric elements: **points**, **lines**, **angles**, and **shapes**, where points serve as the most basic, indivisible concept underlying all geometric relationships.
>
> We first identified key geometric relationships, including **line-based** (e.g., interactions between points and lines), **angle-based** (e.g., angular orientation and symmetry), and **shape-based** (e.g., positions of points or lines relative to a central shape). Using these, we manually created a base dataset of simple problems, expanding it slightly with LLM-generated examples. These problems typically require no additional reasoning, forming the **basic difficulty level**.
>
> To construct the **advanced difficulty level**, we combined base relationships into reasoning chains (2–3 steps) to create more complex problems. This was primarily done using rule-based generation, with manual validation to ensure quality.
>
> Finally, we applied two augmentation methods for diversity: (1) **Adding unrelated information**, where non-essential details were added (some generated using LLMs); and (2) **Re-labeling points**, which used rules to alter point labels without changing problem logic. Both methods preserved the original difficulty level since reasoning chain complexity remained unchanged.
>
>
> >W2 & Q2: Using the LLaMA3-8B-Instruct model for fine-tuning is unconventional and suggesting providing results from fine-tuning the LLaMA3-8B base model.
>
> Thank you for pointing this out. Since our former evaluation focused on instruction-tuned models, we fine-tuned the LLaMA3-8B-Instruct model for consistency. We have now added results from fine-tuning on the LLaMA-3-8B base model, as shown below:
>
> |Model|Line-based (basic)|Line-based (advanced)|Angle-based (basic)|Angle-based (advanced)|Shape-based (basic)|Shape-based (advanced)|
> |-|-|-|-|-|-|-|
> |LLaMA3-8B-Instruct|63.16|42.11|52.17|13.04|26.92|38.46|
> |FinetunedLLaMA3-8B-Instruct|34.14|40.56|32.50|13.75|39.87|40.51|
> |FinetunedLLaMA3-8B-Base|**86.86**|**71.67**|63.16|**54.90**|**93.75**|**61.69**|
> |GPT-4o|77.87|52.91|**66.67**|29.00|87.04|53.38|
>
> Fine-tuning on the base model proves highly effective, significantly improving accuracy across all subsets, and surpassing the best-performing model (GPT-4o) in most categories. Additionally, comparing the two fine-tuned models reveals that the Finetuned LLaMA3-8B-Instruct model underperforms due to frequently selecting the uncertain option "cannot be inferred."
>
> > W3: Including additional experimental metrics commonly used in math domains, such as pass@k, majority voting, or best-of-n, could improve the depth of analysis and provide a more comprehensive evaluation of model performance in this paper.
>
> Thank you for the suggestion. In our main experiments, we used single greedy decoding (temperature set to 0) for all large models. To address your point, we conducted additional experiments on the GPT-3.5-turbo model using multiple sampling runs (temperature set to 1). We evaluated its performance using **pass@k** and **self-consistency with majority voting over k generations** (denoted as acc(SC=k)). The results are as follows:
>
> |k|pass@k (lines-basic)|acc(SC=k)(lines-basic)|pass@k (lines-advanced)|acc(SC=k)(lines-advanced)|pass@k (angles-basic)|acc(SC=k)(angles-basic)|pass@k (angles-advanced)|acc(SC=k)(angles-advanced)|pass@k (shape-basic)|acc(SC=k)(shape-basic)|pass@k (shape-advanced)|acc(SC=k)(shape-advanced)|
> |-----|----------------------|------------------------|-------------------------|---------------------------|-----------------------|-------------------------|--------------------------|----------------------------|-----------------------|-------------------------|--------------------------|----------------------------|
> |1|64.29|64.29|42.46|42.46|42.11|42.11|22.41|22.41|68.75|68.75|38.31|38.31|
> |3|71.43|60.71|62.30|45.24|63.16|52.63|37.93|22.41|93.75|75.00|59.74|38.96|
> |5|75.00|64.29|72.22|44.44|68.42|42.11|55.17|25.86|100.00|75.00|68.83|43.51|
> |10|75.00|64.29|83.33|45.24|84.21|57.89|70.69|32.76|100.00|75.00|77.92|42.21|
>
> We observed that **pass@k** increases rapidly with larger k, indicating a growing likelihood of generating the correct answer through multiple attempts. However, the **majority voting accuracy** (acc(SC=k)) does not improve significantly, suggesting that the model's probability of generating correct answers remains low and that its reasoning over geometric tasks is inconsistent, resulting in dispersed outputs.

---

> > ### Author Response · Authors · 2024-11-24
> > **Look forward to feedbacks**
> >
> > Dear Reviewer XLWM,
> >
> > We first thank you again for your constructive comments. We have addressed your concerns one by one and supplemented more detailed experimental results and explanations. We look forward to further discussion with you and your positive feedback about our rebuttal.
> >
> > Best regards,
> >
> > Authors

---

### Author Response · Authors · 2024-11-26

We would like to sincerely thank the reviewers for their careful evaluation of our paper and for their thoughtful and constructive feedback. The insights provided have been invaluable in improving the clarity and depth of our work. In response to the suggestions, we have made several revisions and additions to the manuscript (marked in red). Specifically, we have included:

1. **Human baselines** for comparison with model performance (Nc3K).
2. A **list of hyperparameters** set for both inference and fine-tuning experiments in Appendix E.1 (Nc3K).
3. **Multi-sampling experiments and analysis** in Appendix E.2 (XLWM).
4. **Additional fine-tuning experiments** and an analysis of the reasons behind the lower performance of the fine-tuned LLaMA3-8B-Instruct model in Appendix E.5 (XLWM, Nc3K, 8Tsz).
5. Experimental results supporting the **analysis of the reasoning path** in Appendix F (8Tsz).

We hope these additions address the reviewers' concerns and enhance the manuscript. We greatly appreciate the reviewers' time and effort in reviewing the paper and look forward to any further discussion or feedback they may have.

---

### Meta-Review · Area_Chair_BMMi · 2024-12-23

**Metareview:**

This paper introduces GeomRel, a novel dataset designed to evaluate LLMs on their ability to understand geometric relationships, addressing limitations in existing datasets that focus solely on final answer accuracy. It proposes the Geometry Chain-of-Thought (GeoCoT) method, which decomposes geometric reasoning into step-by-step processes, significantly improving performance. Strengths include the dataset’s focus on isolating geometric relationship identification, extensive evaluations across multiple models and prompting techniques, and the effectiveness of GeoCoT in enhancing reasoning. However, weaknesses include limited exploration of 3D relationships, scalability issues due to manual dataset curation, narrow focus on isolated geometric tasks, and marginal improvements from fine-tuning in certain cases. All the reviewers lean towards acceptance of the paper with scores 6,6,8. I agree with the reviewers that the strengths of this paper outweigh the

**Additional Comments On Reviewer Discussion:**

Reviewers praised the paper for introducing a valuable dataset (GeomRel), proposing the effective GeoCoT method, and conducting extensive evaluations, but raised concerns about scalability, dataset scope (limited to 2D relationships), unclear fine-tuning results, and insufficient exploration of human baselines. In response, the authors clarified dataset construction, added fine-tuning results on additional models (e.g., LLaMA3-8B-Base and Qwen2-7B), conducted human evaluations for comparison, and provided detailed explanations about key limitations, such as focusing on 2D relationships for foundational reasoning and plans for future scalability improvements. These responses addressed most reviewer concerns. I think this paper is a valuable addon to the community by introducing new evaluations and there are no significant flaws of the paper.

---

### Decision · Program_Chairs · 2025-01-22

Accept (Poster)